# Preventive and Therapeutic Effects of Sericin-Derived Oligopeptides (SDOs) from Yellow Silk Cocoons on Blood Pressure Lowering in L-NAME-Induced Hypertensive Rats

**DOI:** 10.3390/foods14071256

**Published:** 2025-04-03

**Authors:** Chainarong Tocharus, Manote Sutheerawattananonda

**Affiliations:** 1Department of Anatomy, Faculty of Medicine, Chiang Mai University, Chiang Mai 50200, Thailand; chainarongt@hotmail.com; 2School of Food Technology, Institute of Agricultural Technology, Suranaree University of Technology, Nakhon Ratchasima 30000, Thailand

**Keywords:** hypertension, sericin, yellow silk, silk protein, oligopeptides, bioactive peptides

## Abstract

Our previous research has shown that SDOs derived from yellow silk cocoons have hypotensive effects on rats in chronic toxicity testing. This study investigated the potential preventative and therapeutic benefits of SDOs on hypertensive rats induced by L-NAME. The experiment involved nine rat groups: (1) normal control, (2) normal + 200 $mg\;{kg}^{-1}$ BW SDOs, (3) hypertensive (HT) control, (4) HT + 50 $mg\;{kg}^{-1}$ BW SDOs, (5) HT + 100 $mg\;{kg}^{-1}$ BW SDOs, (6) HT + 200 $mg\;{kg}^{-1}$ BW SDOs, (7) HT + enalapril (Ena), (8) HT + soy protein isolate (SPI), and (9) HT + bovine serum albumin (BSA). In the preventative approach, rats received 40 $mg\;{kg}^{-1}$ of L-NAME with the studied substances during the four-week investigation. SDOs given at doses of 100 and 200 $mg\;{kg}^{-1}$ BW demonstrated a significant decrease in systolic blood pressure (SBP) without affecting heart rate (HR). In therapeutic studies, 40 $mg\;{kg}^{-1}$ BW of L-NAME increased SBP in the experimental groups over the first four weeks, resulting in mean SBP values above 150 mmHg. Administering 100 and 200 $mg\;{kg}^{-1}$ BW SDOs and 100 $mg\;{kg}^{-1}$ BW SPI significantly reduced SBP. However, SDOs at 200 $mg\;{kg}^{-1}$ BW exhibited SBP closer to the enalapril group. In functional vascular tests, rats given SDOs at a dose of 200 $mg\;{kg}^{-1}$ BW had the highest relaxation and lowest contraction percentages, like the normal control groups. The research found that SDOs may inhibit and manage hypertension in both healthy and hypertensive rats by safeguarding endothelial cells.

## 1. Introduction

Hypertension is a major global risk factor for morbidity and mortality; it is considered one of the most significant risk factors for various cardiovascular diseases, including coronary artery disease, left ventricular hypertrophy, and valvular heart disease [1]. The 2017 American College of Cardiology–American Heart Association (ACC–AHA) Hypertension Guideline defined hypertension as a systolic BP of 130 mmHg or greater or a diastolic BP of 80 mmHg or greater [2]. Maintaining blood pressure below 130/80 mmHg is recommended for patients with coronary heart disease, congestive heart failure, post-renal transplantation, diabetes mellitus, and stroke [3]. In the United States, the prevalence of hypertension is 45.6%, with a control rate of 61.0% for those treated with a target of less than 140/90 mmHg and 46.6% for those with a target of less than 130/80 mmHg [4]. However, under antihypertensive medications, approximately 10 to 15% of individuals with resistant hypertension still experience uncontrolled blood pressure. Conversely, some experts argue that the focus on strict blood pressure targets may overlook the individual variability in patient responses to treatment. Additionally, lifestyle modifications and alternative therapies could offer effective management strategies for many patients [5].

For long-term management to maintain normal blood pressure, a holistic approach has gained increased acceptability due to the side effects of antihypertensive drugs, particularly the bioactivities of dietary protein and peptides [6,7]. Soy protein is among the dietary proteins that have been reported to exhibit hypotensive properties [8,9]. Known for its high nutritional value and functional qualities, SPI is a commonly consumed dietary protein derived from soybeans. As a complete protein, SPI provides all essential amino acids and serves as a common source of plant-based protein in human diets [10]. Polyunsaturated fatty acids, lecithin, and stigmasterol, abundant in soy, have demonstrated the capacity to safeguard cardiovascular health [11]. Due to their anti-inflammatory and antioxidant qualities, soy protein and flavonoids—two significant components of soy active ingredients—have been shown to lower the risk of cardiovascular diseases (CVDs) [12]. Washburn et al. (1999) reported blood pressure abnormality could be enhanced by the intake of 40 g of soy protein, which contains 68 mg of phytoestrogens [13,14]. It is known that aging-related increases in blood pressure are associated with enhanced arterial stiffness, which could be alleviated by the consumption of phytoestrogens [15]. Additionally, soy intake has been demonstrated to reduce blood pressure, potentially through its natriuretic impact, similar to the diuretic furosemide [16]. Oxidative stress and inflammation have been shown to play a key role in the development of hypertension, suggesting that soy isoflavones might lower blood pressure through their antioxidant and anti-inflammatory effects [13]. BSA is an extensively characterized standard protein frequently used in medical research to assess protein metabolism and bioactivity [17,18]. BSA commonly serves as a reference protein in the development of novel immunochemical assays and can be ingested as an edible protein [19]. In this study, SPI and BSA were included as control protein groups to compare with SDOs. By evaluating the effects of SDOs in relation to these established protein sources, this study could yield valuable insights and possible mechanisms for their blood pressure-lowering ability.

Research indicates that sericin and SDOs derived from yellow silk cocoons have several health advantages, including hypoglycemia, hypotension, cholesterol reduction, immunomodulation, colon cancer prevention, and oxidative stress mitigation [20,21,22,23,24,25,26]. SDOs had ACE inhibitory activities slightly reduced after post-simulated gastrointestinal digestion but significantly increased after in vitro blood plasmin digestion [27]. Comprehensive characterization analyses and antioxidant activities of SDOs derived from yellow silk cocoons have been published, revealing numerous new peptide sequences with the highest ACE inhibitory efficacy [28]. In addition to their hypotensive effects via the ACE inhibition mechanism, SDOs exhibited other pathways for reducing blood pressure in both ex vivo and in vivo settings. The ex vivo investigation indicated that SDOs reduced blood pressure via mechanisms that included smooth muscle vasodilation and calcium ion channel inhibition [29]. This phenomenon may result from the opening or closing of calcium channels, thereby reducing calcium influx and inducing muscular relaxation. SDOs have shown a substantial reduction in phenylephrine (PE)-induced contraction of vascular smooth muscle, mediated by the release of calcium from cellular reserves [29]. Research on oral toxicity in male and female rats indicated that SDOs had hypotensive effects, even at low doses of 50 $mg\;{kg}^{-1}$ BW [30]. SBP started to drop in the second month of the study and remained between 110 and 115 mmHg for the duration of the six-month timeframe. The control groups demonstrated an increase in SBP, surpassing 140 mmHg by the end of the study [30].

Considering the previously discussed health advantages of SDOs from yellow silk cocoons, our previous research has shown their simultaneous effects on multiple organs [22,26,29,30,31]. This may improve overall health and reduce the risk of complications linked to unstable hypertension, particularly as the world aging population increases. Upon legal approval, individuals may include SDOs derived from the enzymatic degradation of organic insect protein into their daily meals. Our recent study has shown that SDOs’ physicochemical characteristics, biological activities, and antioxidant qualities remain stable throughout a six-month storage period at 45 °C [28]. They exhibit water solubility throughout a wide pH spectrum, allowing for incorporation into diverse food products [28]. Moreover, the absence of bitterness and the mild taste of SDOs make them palatable to a diverse array of customers. SDOs are generated from sericin extracted with hot water from cocoons of yellow silkworms, which were exclusively fed with organic mulberry leaves for 24 days. This material undergoes enzymatic hydrolysis and is then spray-dried into powder, achieving an 85% yield [28], thus providing a cost-effective source of novel insect protein peptides.

SDOs derived from yellow silk cocoons have been demonstrated to reduce blood pressure in normal-tensioned rats following long-term oral administration during chronic toxicity testing. However, researchers have yet to explore their ability to prevent and treat high blood pressure in rats, as well as their effects on vascular function under hypertensive conditions. The purpose of this research was to investigate how SDOs reduce blood pressure and influence the way blood vessels function when L-NAME induces high blood pressure. It also investigated how SDOs compared to Ena, SPI, and BSA as preventive and therapeutic agents. As part of the preventive study, rats were given L-NAME orally in addition to the tested substances. This would lead their blood pressure to increase during the testing period. To demonstrate therapeutic effectiveness, rats were given L-NAME during the first phase to induce hypertension. SDOs and tested substances were subsequently given throughout the study period. The blood arteries of the thoracic aorta were evaluated to observe how they relaxed and constricted when acetylcholine (ACh), sodium nitroprusside (SNP), and phenylephrine (PE) were administered. The findings of this research may cast light on the advantages of SDOs for hypertension, as well as their possible action paths via endothelial cells (ECs) and smooth muscle cells (SMCs).

## 2. Materials and Methods

### 2.1. SDOs Preparation

SDOs were produced by the solubilization of sericin in yellow silk cocoons using autoclaving for 30 min [32]. The cocoons were then separated using cheesecloth and enzymatically hydrolyzed employing protease (16 U/g, EC no. 2327522, Sigma, St. Louis, MO, USA). A protease enzyme solution containing 0.01 unit/mL of protease in a 0.036 M calcium chloride (CaCl_2_) solution at a 1:1 volumetric ratio was combined with 300 mL of sericin solution, stirred for 1 h at 37 °C, and then heated to 90 °C for 15 min. The mixture was then cooled and centrifuged at 9500× *g* for 15 min at 4 °C to isolate the solids. The technique used a hollow fiber membrane with a molecular weight cut-off (MWCO) of 5000 to separate oligopeptides under 5 kDa from larger oligopeptides by ultra-membrane filtration (GE Healthcare Bio-Sciences AB, Uppsala, Sweden). The SDOs solution was lyophilized and then kept at ambient temperature until used. Daily preparations for animal testing included combining SDOs powder with distilled water (DW) to provide the amounts required for oral administration. The specifications of SDOs obtained from yellow silk cocoons used in this investigation were previously outlined in Tocharus et al. (2024) [30]. SDOs’ bioactivities and comprehensive characteristics were previously published and available in Sangsawad et al. (2022) [27] and Sangsawad et al. (2025) [28], respectively.

### 2.2. Experimental Animals

The National Laboratory Animal Center, Salaya, Mahidol University, provided us with 108 male Wistar rats that were eight weeks old and weighed between 180 and 200 g. The lab rats were maintained in a room with a light–dark cycle of 12:12 h at a temperature of 25 ± 1 °C, with lighting from 6:00 a.m. to 6:00 p.m. The experimental rats were provided with sufficient standard rodent chow (C.P. Company, Bangkok, Thailand) and deionized (DI) water for the period of the experiment. The National Research Council of Thailand’s Code of Ethics for the Use of Animals in Scientific Research guided the protocols for using experimental animals. The Animal Ethics Committee approved the experimental procedure in accordance with Chiang Mai University’s guidelines for the care and use of laboratory animals under the specified permit number 17/2555.

### 2.3. Hypotensive Effect of SDOs

#### 2.3.1. Preventive Effects of SDOs on the Development of Hypertension

Figure 1 illustrates the research design for the preventive effect of SDOs on hypertension, modified from a study conducted by [33,34]. Nine groups, each consisting of six rats, formed from the experimental animals: 1. normal control received DW (Ctr); 2. normal rats administered 200 $mg\;{kg}^{-1}$ BW SDOs (Ctr + SDOs); 3. control hypertensive (HT) rats induced with 40 $mg\;{kg}^{-1}$ BW L-NAME; 4. HT rats treated with 50 $mg\;{kg}^{-1}$ BW SDOs (administered alongside 40 $mg\;{kg}^{-1}$ BW L-NAME) (HT + 50 $mg\;{kg}^{-1}$ BW SDOs); 5. HT rats treated with 100 $mg\;{kg}^{-1}$ BW SDOs (administered alongside 40 $mg\;{kg}^{-1}$ BW L-NAME) (HT + 100 $mg\;{kg}^{-1}$ BW SDOs); 6. HT rats treated with 200 $mg\;{kg}^{-1}$ BW SDOs (administered alongside 40 $mg\;{kg}^{-1}$ BW L-NAME) (HT + 200 $mg\;{kg}^{-1}$ BW SDOs); 7. HT rats treated with 10 $mg\;{kg}^{-1}$ BW enalapril (administered alongside 40 $mg\;{kg}^{-1}$ BW L-NAME as the positive control group) (HT + Ena); 8. HT rats treated with 100 $mg\;{kg}^{-1}$ BW SPI (administered alongside 40 $mg\;{kg}^{-1}$ BW L-NAME) (HT + SPI); 9. HT rats treated with 100 $mg\;{kg}^{-1}$ BW BSA (administered alongside 40 $mg\;{kg}^{-1}$ BW L-NAME as the negative control group) (HT + BSA). The duration of the preventative testing phase was 4 weeks. We used the tail-cuff technique with an LE5001 non-invasive blood pressure meter (Panlap, Harvard Apparatus, Barcelona, Spain) to assess systolic blood pressure (SBP) and heart rate (HR) during the experiment [35]. We restrained the rats in a heating chamber (PMB-1030 Preheat & Measuring Box, Muromachi Kikai Co., Ltd., Tokyo, Japan) at 37 °C for 15 min to assess SBP [36]. At the conclusion of the experiment, we gave an intraperitoneal injection of pentobarbital sodium (Nembutal) at a dosage of 50 $mg\;{kg}^{-1}$ BW to anesthetize the rats. Upon the rats attaining full anesthesia, we incised their thoracic cavities and promptly excised their hearts to avert any distress to the experimental subjects. We assessed the vascular function of the experimental animals’ blood arteries using an organ bath.

#### 2.3.2. Therapeutic Effects of SDOs After Hypertension Induction

Figure 2 depicts the experimental framework for investigating the therapeutic effects of SDOs, similar to the investigation of their preventative benefits. The rats were categorized into nine groups, each including six animals: 1. control group (normal rats given DW) (Ctr); 2. control + 200 $mg\;{kg}^{-1}$ BW SDOs (normal rats given DW mixed with SDOs at a dosage of 200 $mg\;{kg}^{-1}$ BW) (Ctr + SDOs); 3. hypertension (a control group of HT rats induced with 40 $mg\;{kg}^{-1}$ BW L-NAME) (HT); 4. HT + 50 $mg\;{kg}^{-1}$ BW SDOs (rats administered 50 $mg\;{kg}^{-1}$ BW SDOs concurrently with 40 $mg\;{kg}^{-1}$ BW L-NAME); 5. HT + 100 $mg\;{kg}^{-1}$ BW SDOs (rats administered 100 $mg\;{kg}^{-1}$ BW SDOs in conjunction with 40 $mg\;{kg}^{-1}$ BW L-NAME); 6. HT + 200 $mg\;{kg}^{-1}$ BW SDOs (rats administered 200 $mg\;{kg}^{-1}$ BW SDOs in conjunction with 40 $mg\;{kg}^{-1}$ BW L-NAME); 7. HT + 10 $mg\;{kg}^{-1}$ BW enalapril (rats administered 10 $mg\;{kg}^{-1}$ BW enalapril in conjunction with 40 $mg\;{kg}^{-1}$ BW L-NAME) (HT + Ena); 8. HT + 100 $mg\;{kg}^{-1}$ BW SPI (rats administered 100 $mg\;{kg}^{-1}$ BW SPI concurrently with 40 $mg\;{kg}^{-1}$ BW L-NAME) (HT + SPI); 9. HT + 100 $mg\;{kg}^{-1}$ BW BSA (rats administered 100 $mg\;{kg}^{-1}$ BSA in conjunction with 40 $mg\;{kg}^{-1}$ BW L-NAME) (HT + BSA). The experiment was carried out over a duration of eight weeks. In the first four weeks, we administered L-NAME to the experimental animals at a dosage of 40 $mg\;{kg}^{-1}$ BW to induce hypertension (SBP > 140 mmHg), a procedure that generally requires around four weeks [37,38]. Following the onset of hypertension, the designated rat groups were administered with SDOs or alternative substances. HR and SBP were monitored and recorded during the course of the study. The rats were anesthetized at the end of the experiment by administering an intraperitoneal injection of pentobarbital sodium (Nembutal) at a dosage of 50 $mg\;{kg}^{-1}$ BW. Following the administration of complete anesthesia, we incised the rats’ thoraxes and promptly excised their hearts to avoid any suffering to the experimental subjects. We evaluated the vascular function of the experimental rats’ blood vessels using the organ bath approach, similar to the method used in the preventive study. All experimental rats were gathered for proper euthanasia at the end of the experiment, in compliance with the guidelines of the Faculty of Medicine, Chiang Mai University.

### 2.4. Functional Vascular Study

At the conclusion of the experiment, we euthanized the animals and removed their blood vessels (thoracic aorta) so that the connective tissue could be removed right away.

The blood arteries were sliced into 3 mm long segments and placed in Krebs’s solution, which included (mM) NaCl 122, KCl 5, and [N-(2-hydroxyethyl) piperazine N’-(2-ethanesulfonic acid)] (HEPES) 10; KH_2_PO_4_ 0.5; NaH_2_PO_4_ 0.5; MgCl_2_ 1; glucose 11; CaCl_2_ 1.8 (pH 7.3); maintained at 37 °C with oxygen, as shown in Figure 3. Two tungsten wires passed through the lumen were used to suspend the aortic ring to a resting tension of 1 g, and then we allowed the blood vessels to self-adjust for about an hour before starting the test. Figure 3 illustrates the organ bath setup: one wire was anchored to the glass hook at the bottom of the organ chamber, while the other was connected to a tissue hook attached to a low-compliance force transducer (Radnoti Glass Instruments, Monrovia, CA, USA) that measured isometric force [39,40]. The transducers were linked to single-channel bridge amplifiers (AD Instruments) before connecting to an analog-to-digital data interface (Powerlab, AD Instruments, Colorado Springs, CO, USA) attached to a computer. Tension variations were recorded utilizing Chart 5.5 or 7.0 (AD Instruments), and the data were stored on digital media for further analysis. This test included three substances—ACh, SNP, and PE—to evaluate the functionality of blood arteries, particularly ECs and SMCs, using the organ bath approach.

Figure 4 depicts the three different action paths of the three substances. ACh influences ECs by inducing the release of calcium (Ca^2+^) ions from the endoplasmic reticulum (ER), thereby elevating Ca^2+^ concentrations. This subsequently alters the activity of endothelial nitric oxide synthase (eNOS) via modifying the phosphorylation of eNOS proteins, leading to the synthesis of nitric oxide (NO) from L-arginine, resulting in SMC relaxation [41]. SNP, a powerful vasodilator used for acute hypertension treatment, produces NO through reduction, which can directly permeate SMCs and facilitate their relaxation. PE, a selective α₁-adrenergic receptor (α_1_-AR) agonist, binds to α₁-AR located on SMCs and activates the Gq protein, leading to the stimulation of phospholipase C (PLC) [42]. PLC breaks down phosphatidylinositol 4,5-bisphosphate (PIP_2_) into inositol 1,4,5-triphosphate (IP_3_) and diacylglycerol (DAG). IP₃ binds to IP₃ receptors (IP_3_R) on the sarcoplasmic reticulum (SR), causing Ca^2+^ release into the cytoplasm, leading to smooth muscle contraction [43]. On the other hand, DAG binds to protein kinase C (PKC) and translocates to the plasma membrane. PKC phosphorylates ion channels that increase Ca^2+^ influx from extracellular fluid, further promoting contraction [44].

### 2.5. Data Analysis

The SBP and HR data were analyzed using two-way ANOVA. For vascular functional data, one-way ANOVA was used. The least significant difference (LSD) test was used to compare the differences between the normal control group and the normal + 200 $mg\;{kg}^{-1}$ BW SDOs group, the normal control group and the HT control group, and the HT control group and the HT groups after the substances were administered. Statistical significance was determined using SPSS 16.0 software (SPSS Inc., Chicago, IL, USA) with confidence levels of *p* < 0.05, 0.01, and 0.001.

## 3. Results and Discussion

### 3.1. Preventive Effects of SDOs During Induced Hypertension

Figure 5A illustrates the hypotensive effects of SDOs derived from yellow silk cocoons and other evaluated substances in both normal and hypertensive rats. Throughout the four-week investigation, the SBP values remained the same for both the normal control rats and the normotensive rats with 200 $mg\;{kg}^{-1}$ BW SDOs, suggesting that SDOs did not lower blood pressure in normotensive rats within the four-week treatment period. This corresponds with our prior research indicating the initial development of hypotensive effects by the eighth week of oral administration [30]. In all rats administered L-NAME, SBP escalated progressively from normal to hypertensive levels during the first and fourth weeks, ultimately reaching 147.20 ± 1.74 mmHg for those receiving only L-NAME. Nonetheless, when the rats were administered L-NAME alongside other tested substances such as Ena, SPI, and SDOs, their SBP levels significantly decreased to be lower than 120 mmHg by the fourth week, approaching those of normotensive rats. SDOs at dosages of 100 and 200 $mg\;{kg}^{-1}$ BW showed a preventive benefit against hypertension in rats, reducing SBP to 117 ± 3.53 and 112.16 ± 2.71 mmHg, respectively. The SBP of rats given L-NAME along with BSA and SDOs at a dose of 50 BW was significantly different from that of the HT control group. However, their SBP values were still above 120 mmHg, with the SDOs group having 133.12 ± 3.46 mmHg. This indicates that while the treatment altered the SBP readings compared to the HT control group, the levels remained elevated, suggesting that the intervention was not fully effective in normalizing blood pressure. The notably high SBP in the HT+ 50 mg kg^−1^ SDOs group highlights the need for a higher dose of SDOs to reach the efficacy of these treatments. The rats had comparable HRs (Figure 5B), indicating that the tested substances had no effect on the HR.

### 3.2. Therapeutic Effects of SDOs After Hypertension Induction

Figure 6A demonstrates the hypotensive effects of all studied substances relative to the control groups. The rats given L-NAME showed complete hypertension, with SBP surpassing 150 mmHg after a period of 4 weeks. The SBP results for L-NAME-induced hypertensive rats in the HT, HT + 50 $mg\;{kg}^{-1}$ BW SDOs, HT + 100 $mg\;{kg}^{-1}$ BW SDOs, HT + 200 $mg\;{kg}^{-1}$ BW SDOs, HT+ Ena, HT+ SPI, and HT+ BSA groups exhibited initial values of 111 ± 1.2, 108.60 ± 4.94, 112.3 ± 1.10, 110.92 ± 1.64, 108.25 ± 2.80, 109.22 ± 1.68, and 106.00 ± 1.12 mmHg, respectively, and subsequently increased to 151.33 ± 1.36, 154.63 ± 2.60, 153.53 ± 0.83, 159.27 ± 3.41, 151.27 ± 1.84, 159.17 ± 1.36, and 157.28 ± 1.25 mmHg, respectively. By the seventh week of the investigation, following three weeks of administering tested substances, the hypotensive effects of some began to show up. Enalapril showed the most pronounced effect on reducing SBP, followed by SDOs at 200 $mg\;{kg}^{-1}$ BW, 100 $mg\;{kg}^{-1}$ BW, and SPI, in that order. In contrast, rats receiving SDOs at 50 $mg\;{kg}^{-1}$ BW exhibited SBP levels similar to the HT group. The SBP levels of the rats that were given enalapril (117.98 ± 2.31 mmHg) became closer to those in the normal control groups in the eighth week of the study, which was four weeks after the substance being tested was given. After that, rats given SDOs at doses of 200 and 100 $mg\;{kg}^{-1}$ BW had SBP readings of 129.01 ± 1.48 mmHg and 137.53 ± 2.34 mmHg, respectively. The next group of rats to be given SPI had an SBP reading of 140.30 ± 3.15 mmHg. BSA exhibited a modest hypotensive effect, showing a significant difference relative to the HT group; nonetheless, SBP measurements remained at hypertensive levels of around 150 mmHg. We found that the hypotensive effects of SDOs in hypertensive rats were dose-dependent, requiring a minimum dosage of 100 $mg\;{kg}^{-1}\;$BW for efficacy and required at least three weeks of administration to observe the significant effect. All substances tested demonstrated no significant impact on the HR for both normotensive and hypertensive rats (Figure 6B).

The preventive and therapeutic experiments revealed a similar pattern: SDOs from yellow silkworm cocoons, given at dosages of 100 and 200 $mg\;{kg}^{-1}$ BW for more than 3 weeks, significantly reduced SBP in HT rats. However, the SBP-lowering effect caused by SDOs was not as strong as Ena throughout the research period. SPI exhibited roughly the same blood pressure-lowering effect as SDOs. Extended periods of administration, an increased dosage of SDOs and SPI, or both may be necessary to improve both efficiency and effectiveness. Notably, SDOs from yellow silk cocoons and SPI are both antioxidant-rich proteins [13,28], which may mitigate oxidative stress damage produced by L-NAME in ECs [42]. Our earlier study discovered that SDOs demonstrated hypotensive efficacy via three possible modes of action: (1) ACE inhibitory activity, (2) calcium channel blockage, and (3) antioxidant activity, presumably via mitochondrial oxidative stress mitigation [27,28,29,45,46].

SDOs’ potential function in controlling blood pressure is highlighted by their suppression of the angiotensin-converting enzyme (ACE). By converting angiotensin I into angiotensin II, a strong vasoconstrictor, ACE plays a crucial part in the renin–angiotensin system [47]. SDOs may lower angiotensin II levels by blocking this enzyme, which could promote vasodilation and lower blood pressure. Furthermore, SDOs demonstrated calcium channel-blocking activity suggests their ability to relax blood vessels and lower blood pressure through this distinct pathway [29]. This mechanism could possibly complement ACE inhibition and provide a multifaceted approach to cardiovascular protection [48,49]. The ability of SDOs to influence intracellular calcium levels may also have implications in reducing vascular stiffness and improving cardiac function. Calcium channel blockage may enhance blood flow and reduce myocardial oxygen demand [50,51], further emphasizing the cardioprotective potential of SDOs. Another mechanism, the antioxidant activity of SDOs, particularly via mitochondrial mitigation, may address oxidative stress at its cellular source. By mitigating oxidative stress within mitochondria, SDOs may protect cells from oxidative damage and preserve mitochondrial integrity. This protective effect can enhance cellular resilience and may also reduce apoptosis and tissue damage from oxidative stress [52].

Endothelial mitochondrial dysfunction plays a crucial role in the development of atherosclerosis. Various risk factors, including high glucose levels, hypertension, ischemia, hypoxia, and diabetes, exacerbating mitochondrial dysfunction in endothelial cells, which lead to increased oxidative stress and impaired vascular function [53]. To counteract these adverse effects, SDOs act as an exogenous antioxidant, may revive the redox state balance of endothelial cells under oxidative stress by directly neutralizing ROS radicals, consequently enhancing the production of endogenous antioxidant enzymes [54]. Importantly, the antioxidant properties of SDOs may synergize with their ACE inhibitory and calcium channel-blocking activities, amplifying their overall preventive and therapeutic potential. Furthermore, after undergoing enzymatic hydrolysis, sericin hydrolysates exhibited even stronger antioxidant properties compared to native sericin [45,46]. This enhancement is likely due to the smaller peptides and free amino acids generated during hydrolysis, which possess greater radical scavenging activity [55,56]. This synergy may, in part, be attributed to specific bioactive components within SDOs, such as peptide sequences with potent ACE inhibitory properties.

### 3.3. Effects of Blood Vessel Function

#### 3.3.1. Preventive Groups

##### Effect of SDOs on Vascular Function by Acetylcholine (ACh)-Induced Relaxation

Figure 7A shows the ACh-induced relaxation curve of blood vessels in response to high levels of ACh in all the groups of rats that were given L-NAME. This includes both the treatment group and the control group. Initially, the relaxation was set at 0% and gradually approached 100% as the concentration of ACh increased from $10^{-12}$ to $10^{-5}\;$M, with a decline initiating about $10^{-9}\;$M. Figure 7B illustrates the maximum relaxation percentage values and significant statistical differences. Based on the results, the rats could be distinctly categorized into four groups. The first group demonstrated the highest degree of relaxation, including the Ctr and Ctr + 200 $mg\;{kg}^{-1}$ BW SDOs. The second group represented the lowest relaxation group, including HT and HT + 50 $mg\;{kg}^{-1}$ BW SDOs. The third group included the high-relaxation cohort, which contained HT + 200 $mg\;{kg}^{-1}$ BW SDOs and HT+ SPI. The fourth group included the median-to-low relaxation group, which contained HT + 100 $mg\;{kg}^{-1}$ BW SDOs, HT+ Ena, and HT+ BSA. L-NAME reduced the vasodilatory capacity of blood vessels by less than one-third, as observed in the HT control group (28.60 ± 4.43%). SDOs at 50 $mg\;{kg}^{-1}$ BW exhibited a vascular relaxation rate of 27.50 ± 5.03%, whereas SDOs at 100 and 200 $mg\;{kg}^{-1}$ BW displayed increased relaxation rates of 54.20 ± 12.40% and 82.00 ± 6.44%, respectively. HT + Ena had a relaxation rate of around 49.00 ± 15.13%, while BSA somewhat enhanced the relaxation rate to 33.00 ± 15.52%. This research demonstrated that SPI at 100 $mg\;{kg}^{-1}$ BW (87% relaxation rate) and SDOs at 200 $mg\;{kg}^{-1}$ BW significantly improved relaxation percentages in HT groups, indicating that SDOs and SPI may help the endothelium-dependent vasorelaxation abilities of blood vessels sustained from the damage caused by L-NAME in the preventive experiment.

The ACh mechanism on blood vessels involved the stimulation of endothelial cells (endothelium-dependent relaxation) [57,58]. Primarily, ACh binds to the muscarinic receptor of the endothelial cell, which triggers eNOS to produce and release NO from the endothelial cells and diffuse into SMCs. This activates the soluble guanylate cyclase (sGC), which converts the guanosine triphosphate (GTP) to cGMP, signaling the smooth muscle to relax [58,59].

L-NAME damages ECs and eNOS via dysfunction of Ca^2+^ signaling and reduction of eNOS’s active form. Endothelial Ca^2+^ signaling is crucial to smooth muscle contraction, especially in sites that contact SMCs since they contain Ca^2+^-dependent effector proteins that modulate the contractile state of smooth muscle. In a hypertension state, local IP_3_-mediated endothelial Ca^2+^ signals occur less frequently and at lower amplitude compared to a normal state [42], reducing the ability of the EC to oppose vascular tone and increasing the contractile state of SMCs [42].

The phosphorylation of eNOS is a key regulatory mechanism that increases NO production, which dilates blood vessels and keeps endothelial function stable [36]. Multiple protein kinases such as Akt, AMPK, and PKA phosphorylate eNOS at Ser1177 (humans) or Ser1179 (rodents), increasing their activity and enhancing NO synthesis [60]. The presence of L-NAME reduces the protein expression of phosphorylated eNOS (p-eNOS), the activated form of eNOS. L-NAME could bind to the L-arginine binding site on eNOS, resulting in inhibition of NO synthesis along with ROS generation [36].

SDOs and SPI have been shown to enhance ACh-induced vasorelaxation. SPI is rich in isoflavones, particularly daidzein and genistein [13,61]. Through antioxidant properties, these compounds have been shown to stimulate NO production via the L-arginine/NO-dependent pathway and the reduction in ROS in endothelial cells [62,63]. This may be associated with the increase in NO production and its signaling on vascular smooth muscle in response to ACh, augmenting the vasorelaxation effect. SPI may also enhance the sensitivity of vascular smooth muscle cells (VSMCs) to NO [64], allowing for more effective relaxation. This mechanism complements its role in boosting NO production and reducing factors that impair NO signaling. Similarly, SDOs may act through a comparable mechanism, with their antioxidant and endothelial-protective effects contributing to improved vascular relaxation in response to ACh.

##### Effect of SDOs on Vascular Function by SNP-Induced Relaxation

Figure 8A demonstrates the relaxation percentages of aortic dilatation induced by SNP at concentrations from $10^{-12}$ to $10^{-6}$ M. The response curves of rats across all groups exhibited variability between $10^{-9}$ and $10^{-7}$ M, with a comparable maximal dilation percentage at $10^{-6}$ M (Figure 8B). SNP spontaneously releases NO when it comes into contact with tissue [65], allowing it to enter the SMC directly, resulting in 100% maximum relaxation percentages in all rat groups. This demonstrated that the SMC was still fully functional. In contrast to ACh treatment, maximum relaxation percentages in vascular activities were found to be significantly different, as they were induced by NO generated via eNOS in ECs [57,59,66]. This indicated that L-NAME had a detrimental effect on mainly ECs and not SMCs [67]. As a result, all of the substances examined in this experiment showed comparable relaxation percentages. This demonstrates the relaxation response of SMC to NO generated by SNP, not NO produced by the EC. The results of the vascular activity test reveal an important contrast between the functions of SMCs and ECs in vascular function. Notably, L-NAME treatment demonstrated that SMCs remained unaffected, suggesting their resilience under these conditions. The lower maximum relaxation percentages observed with ACh treatment, on the other hand, point to a problem in ECs rather than SMCs. This pathway is demonstrated by the relaxation rates of the hypertensive control group, which did not significantly differ from those of the other groups in the study.

##### Effect of SDOs on Vascular Function by PE-Induced Contraction

Figure 9A illustrates the percentage contraction of blood vessels in response to an increased concentration of PE, ranging from $10^{-9}$ to $10^{-6}$ M. The graph indicated that the contraction curves could be categorized into three different groups: the control HT group exhibited the highest level of contraction, the HT + SDOs at 200 $mg\;{kg}^{-1}$ BW group had the lowest contraction, while the other study groups were within the intermediate range of contractions. Induced by PE, L-NAME significantly affected vasoconstriction, with the HT-group rats showing the highest percentage contraction curve, whilst the HT + SDOs at 200 $mg\;{kg}^{-1}$ BW demonstrated the lowest response. Figure 9B depicts the maximum contraction percentages for all studied groups. The Ctr and Ctr+ SDOs had comparable maximum contraction rates of 58.33 ± 2.67% and 66.00 ± 4.06%, respectively; however, the HT control group demonstrated the highest maximum contraction rate at 82.60 ± 4.97%. The rats in the HT+ Ena, HT+ SPI, and HT+ BSA groups had comparable maximum contraction rates. The HT rats administered SDOs at doses of 50, 100, and 200 $mg\;{kg}^{-1}$ BW exhibited significantly reduced maximum vasoconstriction rates of 58.20 ± 3.20%, 61.80 ± 6.16%, and 39.20 ± 4.16%, respectively, in comparison to the HT-group rats. SDOs efficiently decreased contraction caused by PE in hypertensive rats, particularly at a dose of 200 $mg\;{kg}^{-1}$ BW.

SDOs significantly reduced contractions induced by PE, suggesting that SDOs may interfere with IP_3_ signaling or intracellular Ca^2+^ dynamics, thereby attenuating smooth muscle contraction [29]. PE is known to stimulate IP_3_-dependent Ca^2+^ release from the intracellular store. Upon binding of IP_3_ to its receptors on the SR membrane, calcium channels open, facilitating the efflux of calcium ions into cytoplasm [68,69]. Thus, SDOs’ vascular effects are likely linked to a decrease in IP_3_-dependent Ca^2+^ releases from SR, which in turn lessens the vasoconstriction brought on by PE.

#### 3.3.2. Therapeutic Groups

##### Effect of SDOs on Vascular Function by ACh-Induced Relaxation

Figure 10A clearly categorizes the percentage relaxation curves of blood vessels induced by Ach into four groups: The highest relaxation percentage included Ctr, Ctr + SDOs, and HT + 200 $mg\;{kg}^{-1}$ BW SDOs. The second highest percentage of relaxation consisted of HT + SPI and HT + 100 $mg\;{kg}^{-1}$ BW SDOs. The middle-range relaxation percentage was made up of HT + BSA and HT + 50 $mg\;{kg}^{-1}$ BW SDOs. The lowest relaxation percentage included both HT and HT + Ena. The HT+ SDOs at 200 $mg\;{kg}^{-1}$ BW had a response curve that was comparable to that of the Ctr and Ctr+ SDOs. This showed that they were effective at restoring vascular function to a near-normal level. Figure 10B similarly illustrates four distinct groups. Among the groups that were studied, Ctr, Ctr + SDOs, and HT + 200 $mg\;{kg}^{-1}$ BW SDOs, SDOs had the highest percentages of maximum relaxation, which were close to 100%. HT + 100 $mg\;{kg}^{-1}$ BW SDOs and HT + SPI had the next highest percentage of maximum relaxation at 92 ± 2.50% and 88.50 ± 2.38%, respectively. HT + 50 $mg\;{kg}^{-1}$ BW SDOs and HT + BSA had middle levels of relaxation, with scores of 69.83 ± 4.28% and 75.25 ± 2.18%, respectively. HT and HT+ Ena had the lowest percentages of maximal relaxation at 41.50 ± 7.31% and 53.80 ± 2.20%, respectively. The effects of SDOs at dosages of 100 and 200 BW on vascular relaxation in hypertensive rats were very similar to those observed in normal control rats (Figure 10A,B), despite considering that the maximum relaxation percentages of the HT with tested substances were found to be significantly different from the HT control, with the exception of the HT + Ena. BSA and SPI contributed to the improvement of vascular function, whereas enalapril had the lowest impact on it among the substances tested. This study showed that enalapril could lower SBP; nevertheless, enalapril does not directly restore eNOS function and NO production, especially in the therapeutic study where the rats were induced with L-NAME for 4 weeks prior to the drug administration. Long-term L-NAME treatment decreased eNOS activity and NO bioavailability, leading to chronic vasoconstriction and increased mechanical stress on the arterial wall [70]. This consequently causes smooth muscle hypertrophic remodeling, which increases vessel wall thickness and stiffness [36]. Despite the reduction in SBP, enalapril may not be able to reverse this remodeling within 4 weeks [71]. Thus, its response to ACh remains weak and unable to improve ACh-induced vasorelaxation, which is eNOS- and NO-mediated relaxation, as effectively as other tested substances.

A previous study reported that BSA could enhance endothelium-dependent vasodilation of many substances, including acetylcholine, carbachol, ADP, ATP, and ionophore [72,73]. ACh-induced vasorelaxation, especially in therapeutic studies, was improved in the presence of BSA. The observed improvement in vasorelaxation was possibly due to the stabilizing ability of BSA on vasodilating agents such as NO [73]. In addition, BSA may improve ACh-induced vasorelaxation similar to SPI and SDOs since it was reported to exhibit antioxidants properties and ACE inhibition [72].

##### Effect of SDOs on Vascular Function by SNP Relaxation

Figure 11A shows the relaxation response curves for aortic dilatation induced by SNP. The relaxation curve resembled the relaxation response curve observed in preventative groups, exhibiting just a little variation in shape between concentrations of $10^{-10}$ M and $10^{-8}$ M of SNP. Figure 11B demonstrates that there was no significant difference in the percentage of maximum aortic dilatation due to SNP across all groups in the treatment phase, which approached 100%. This suggests that SMCs in both normal and HT groups were effectively and equally able to relax in response to NO produced by SNP, indicating that SMCs in both groups were fully functioning. SNP functions as a direct donor of nitric oxide (NO), which activates soluble guanylate cyclase (sGC) in VSMCs, increasing cyclic GMP (cGMP) levels and causing relaxation [65]. This pathway bypasses the ECs entirely, making it an effective tool for assessing SMC function independently of EC influences. Given that SNP-induced relaxation does not require EC involvement, the influence of the tested compounds did not significantly differ in the relaxation response. This indicates that L-NAME had a more negative impact on ECs than SMCs [67].

##### Effect of SDOs on Vascular Function by PE-Induced Contraction

Figure 12A shows that HT rats exhibited the highest contraction response to PE, whereas HT + 200 $mg\;{kg}^{-1}$ BW SDOs had the lowest. The curves show four groups of contraction responses: (1) HT control, HT + Ena, and HT + 50 $mg\;{kg}^{-1}$ BW SDOs; (2) HT + 100 $mg\;{kg}^{-1}$ BW SDOs and normal Ctr rats; (3) HT + BSA and Ctr + SDOs; and (4) HT + SPI and HT + 200 $mg\;{kg}^{-1}$ BW SDOs. In Figure 12B, the maximum contraction percentages of all studied groups are shown in descending order: HT (80.67 ± 4.92%), HT + Ena, HT + 50 $mg\;{kg}^{-1}$ BW SDOs (69.80 ± 5.88%), Ctr (60.17 ± 6.53%), HT + 100 $mg\;{kg}^{-1}$ BW SDOs (60.83 ± 3.43%), Ctr + SDOs (51.52 ± 6.88%), HT + BSA (52.15 ± 6.50%), HT + SPI (41.12 ± 4.20%), and HT + 200 $mg\;{kg}^{-1}$ BW SDOs (37.50 ± 4.28%). When comparing the maximum percentage of blood vessel contraction caused by PE among the HT treatment groups, significant differences were found in HT + BSA (*p* < 0.05) and HT + SPI and HT + 200 $mg\;{kg}^{-1}$ BW SDOs (*p* < 0.001), as illustrated in Figure 12B. The findings demonstrated that BSA, SPI, and SDOs had a positive effect on lowering the vascular activity of SMCs in HT rats induced by PE.

According to the results of this investigation, SDOs showed both protective and therapeutic effects by reducing SBP without affecting HR and maintaining SMC relaxation and contraction within normal ranges when L-NAME was present. Because SDOs are oligopeptides with strong antioxidant activities and have previously been reported to have high ACE inhibitory activity [27,28], this study demonstrated that another possible pathway for SDOs’ blood pressure-lowering effect could be through the improvement of vascular function by promoting blood vessel relaxation via ECs when induced by ACh, a substance that interacts with ECs to convert eNOS into active form [36]. The study demonstrated that SDOs may protect ECs from L-NAME damage, improving blood artery function in HT rats and decreasing SMC contraction activity when treated with PE, demonstrating a significant vasodilation impact via the endothelium and vascular smooth muscle.

The therapeutic study revealed that BSA could alleviate the PE-induced vasocontraction. With its antioxidant properties, BSA may act similarly to SDOs and SPI but less pronounced, possibly due to low dosage. Additionally, a previous study reported that BSA could bind to NO and exert vasorelaxation on precontracted rabbit aortic rings with norepinephrine [74,75]. The study further demonstrated that the free sulfhydryl group of serum albumin, which is found at cysteine 34, could combine with NO to create a stable S-nitrosothiol that can relax blood vessels and stop platelets from forming [74]. This allows protein-bound sulfhydryl species to participate in the action and metabolism of NO by prolonging its half-life and preserving its bioactivities.

Like SDOs, SPI efficiently attenuated PE-induced contraction in the isolated aorta, indicating that it possesses vasorelaxant effects. SPI is comprised of bioactive peptides with known cardiovascular benefits, such as antihypertensive and vasorelaxant effects [64]. One probable mechanism for SPI’s vasorelaxant effects is NO generation, which enhances endothelial function and inhibits vasoconstrictive pathways [76,77]. SPI may increase NO production by activating eNOS, resulting in vasodilation and a decreased contractile response to PE. Furthermore, SPI and soy protein hydrolysate-like SODs can function as ACE inhibitors, lowering Ang II generation and reducing vasoconstriction [78,79]. A previous study indicates that soy-derived peptides may interfere with calcium influx in VSMCs, lowering intracellular calcium levels and resulting in reduced contractility [80]. Furthermore, the antioxidant characteristics of SDOs and SPI may help protect endothelial cells from oxidative stress, which promotes vasorelaxation [81,82].

## 4. Conclusions

This research demonstrated the preventive and therapeutic potential of SDOs de-rived from yellow silk cocoons in hypertensive rats. SDOs at 200 mg kg^−1^ BW significantly lowered SBP in L-NAME-induced hypertensive rats, comparable to enalapril. SDOs had no negative impact on SBP or HR in normotensive rats, demonstrating their safety in such conditions. The investigation on vascular function reveals that SDOs at 200 mg kg^−1^ BW considerably maintained vascular relaxation and contraction near to normal by protecting ECs. This may be due to their ACE inhibitory and antioxidant properties, resulting in decreased SBP. Our findings suggest that SDOs may be effective as bioactive peptides for decreasing the risk of high blood pressure. It also shows potential for their application in functional meals or nutraceuticals designed to promote heart health. These findings suggest that including SDOs in dietary regimens might be a strategic approach to controlling hypertension.

## Figures and Tables

**Figure 1 foods-14-01256-f001:**
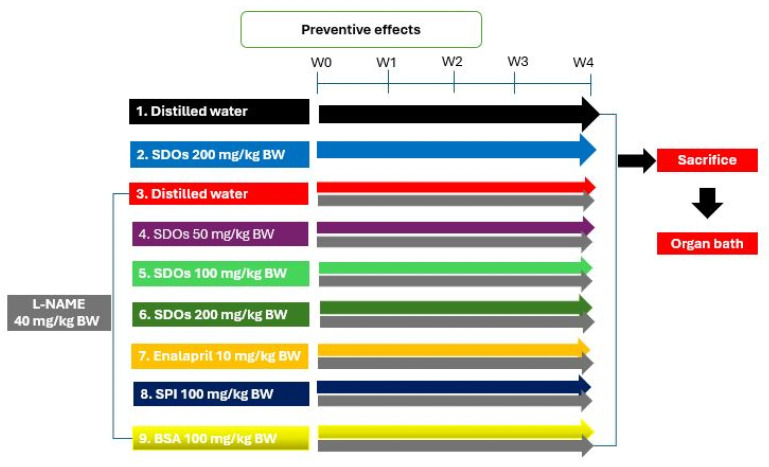
Preventive effects of SDOs experimental framework.

**Figure 2 foods-14-01256-f002:**
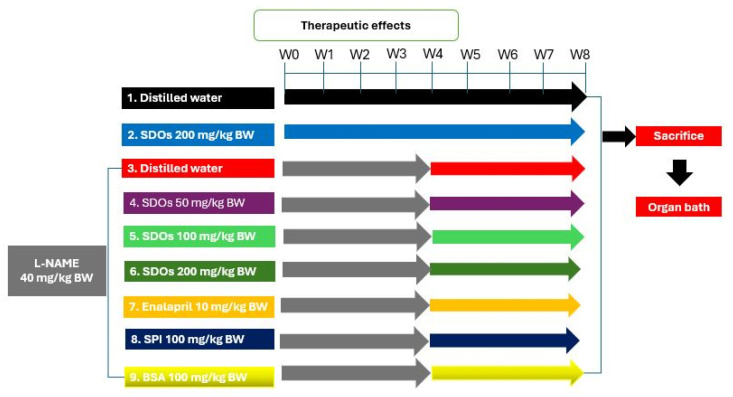
Therapeutic effects of SDOs experimental framework.

**Figure 3 foods-14-01256-f003:**
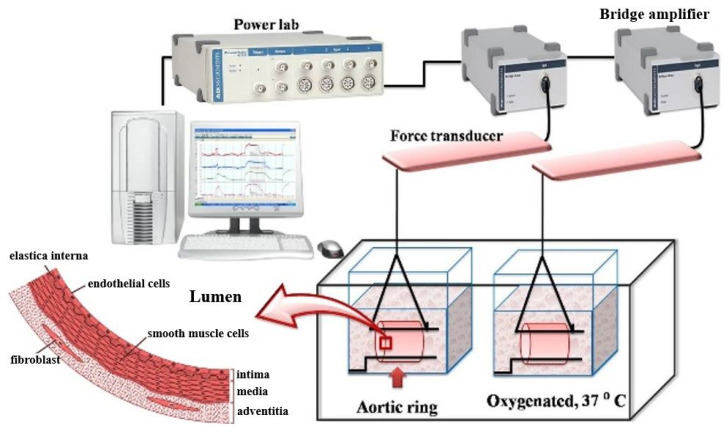
Organ bath setup for functional vascular study.

**Figure 4 foods-14-01256-f004:**
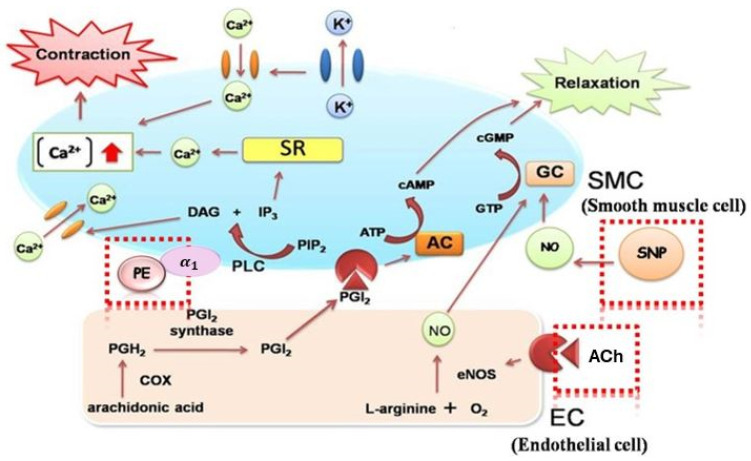
Sites of action for acetylcholine (ACh), sodium nitroprusside (SNP), and phenylephrine (PE) for vascular testing.

**Figure 5 foods-14-01256-f005:**
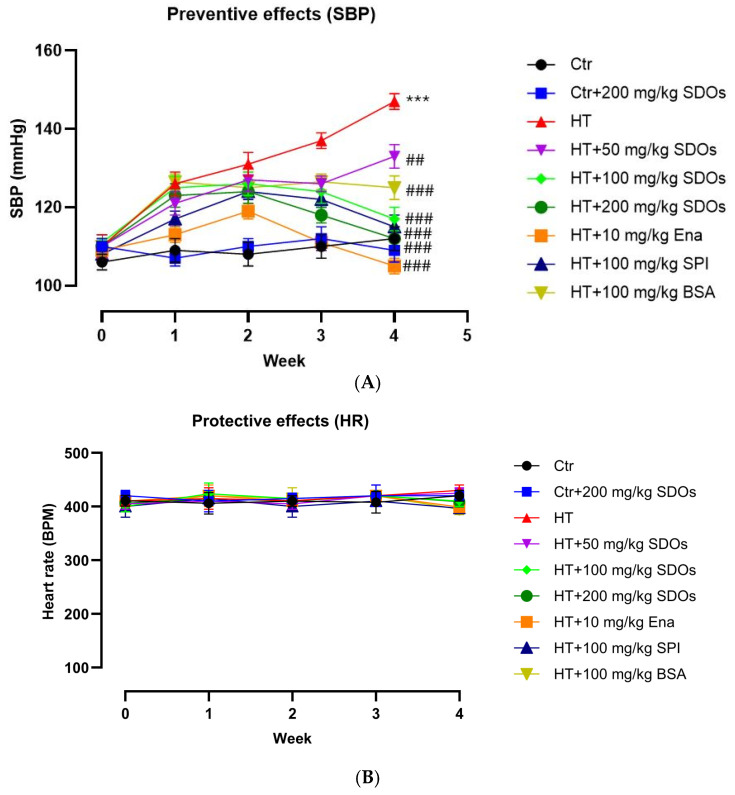
The effect of SDOs on (**A**) systolic blood pressure and (**B**) heart rate during the preventive test. *** *p* < 0.001 compared to the control group (Ctr). ## *p* < 0.01, ### *p* < 0.001 compared to the hypertensive group (HT).

**Figure 6 foods-14-01256-f006:**
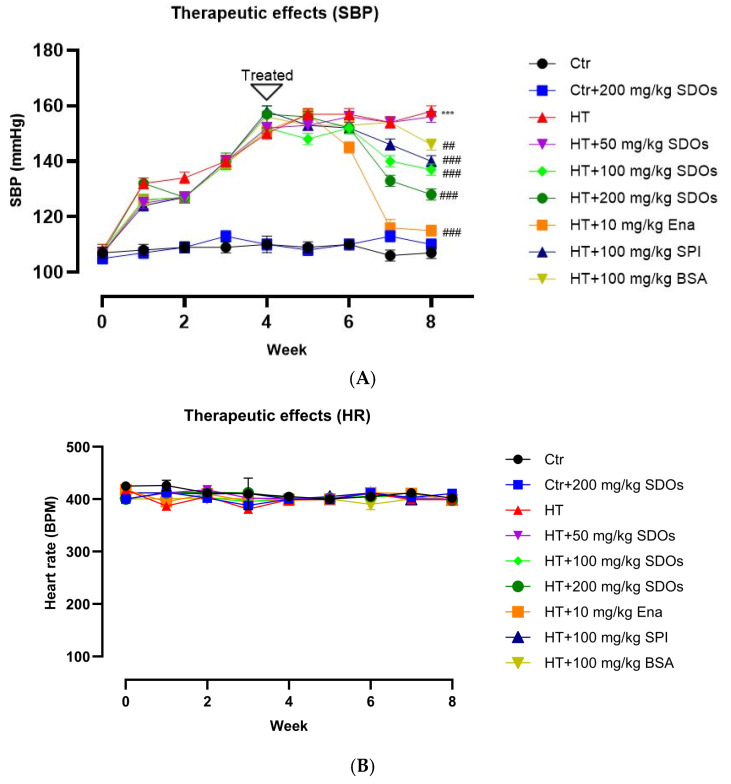
The effect of SDOs on (**A**) systolic blood pressure and (**B**) heart rate during the therapeutic test. *** *p* < 0.001 compared to the control group (Ctr). ## *p* < 0.01, ### *p* < 0.001 compared to the hypertensive group (HT).

**Figure 7 foods-14-01256-f007:**
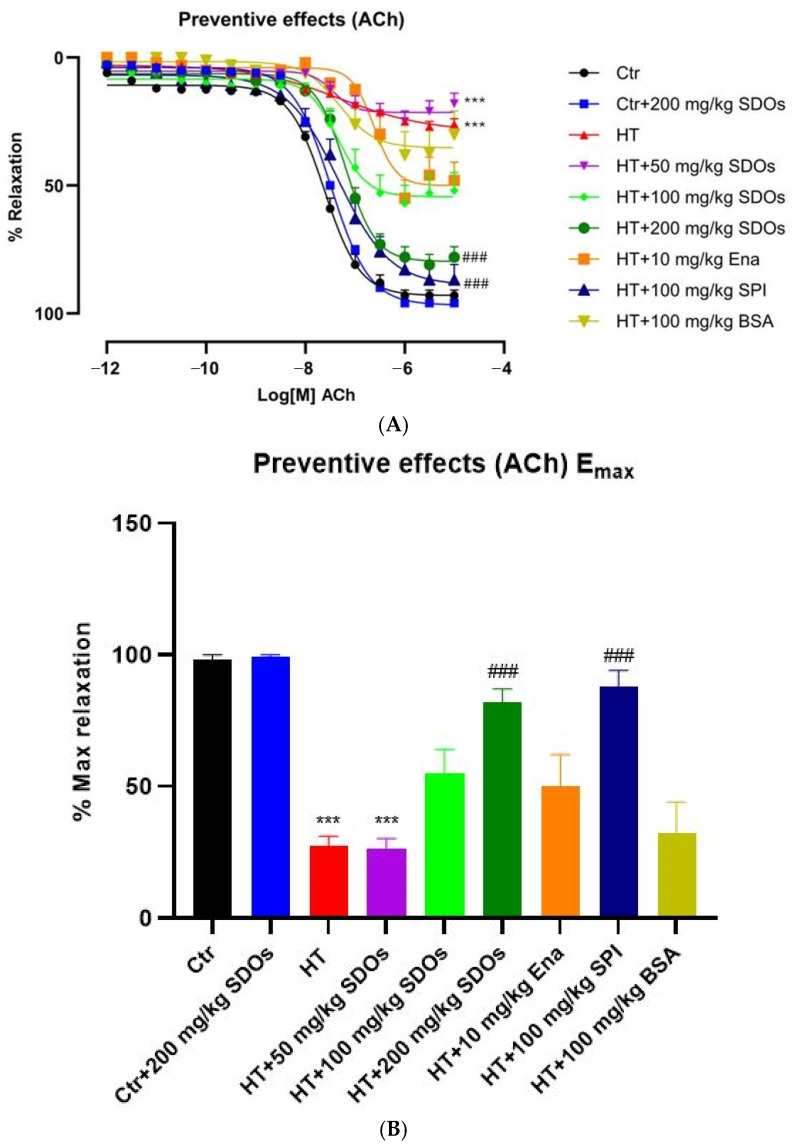
The effects of SDOs on aortic relaxation induced with ACh (**A**) and the maximum relaxation percentage (**B**) obtained from graph A during the preventive test. *** *p* < 0.001 compared to the control group (Ctr). ### *p* < 0.001 compared to the hypertensive group (HT).

**Figure 8 foods-14-01256-f008:**
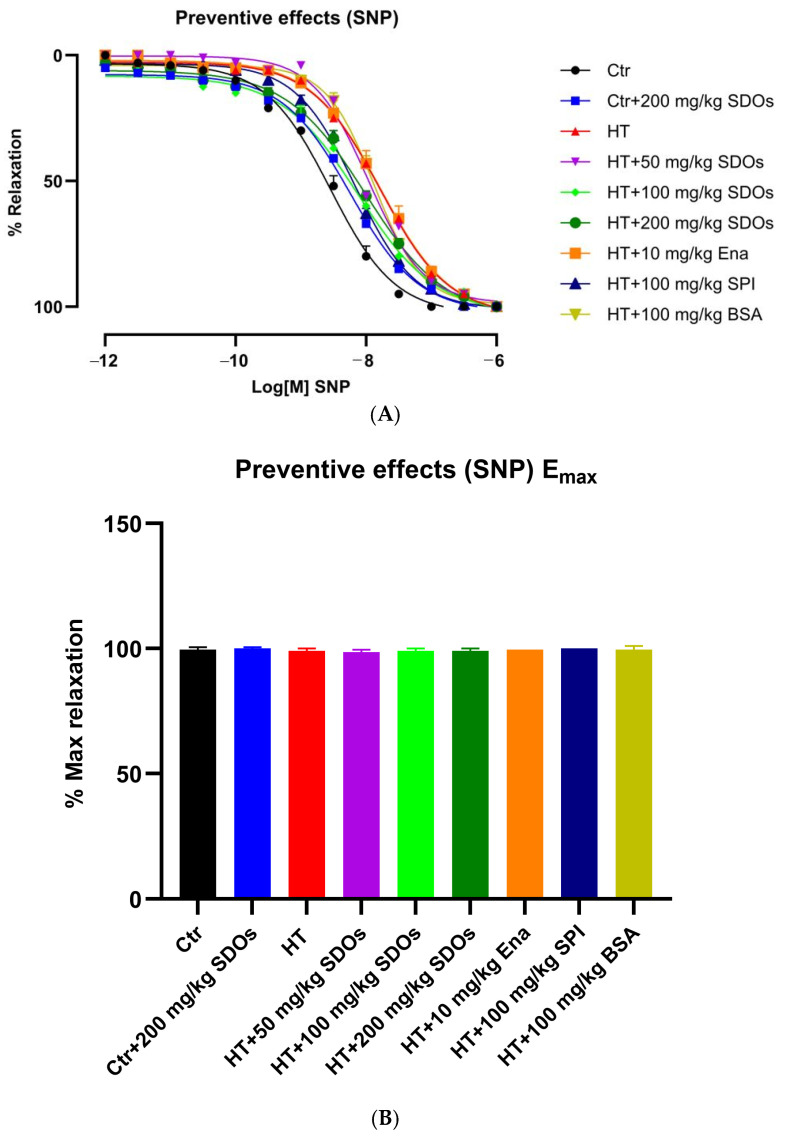
The effects of SDOs on aortic relaxation induced with SNP (**A**) and the maximum relaxation percentage (**B**) obtained from graph A during the preventive test.

**Figure 9 foods-14-01256-f009:**
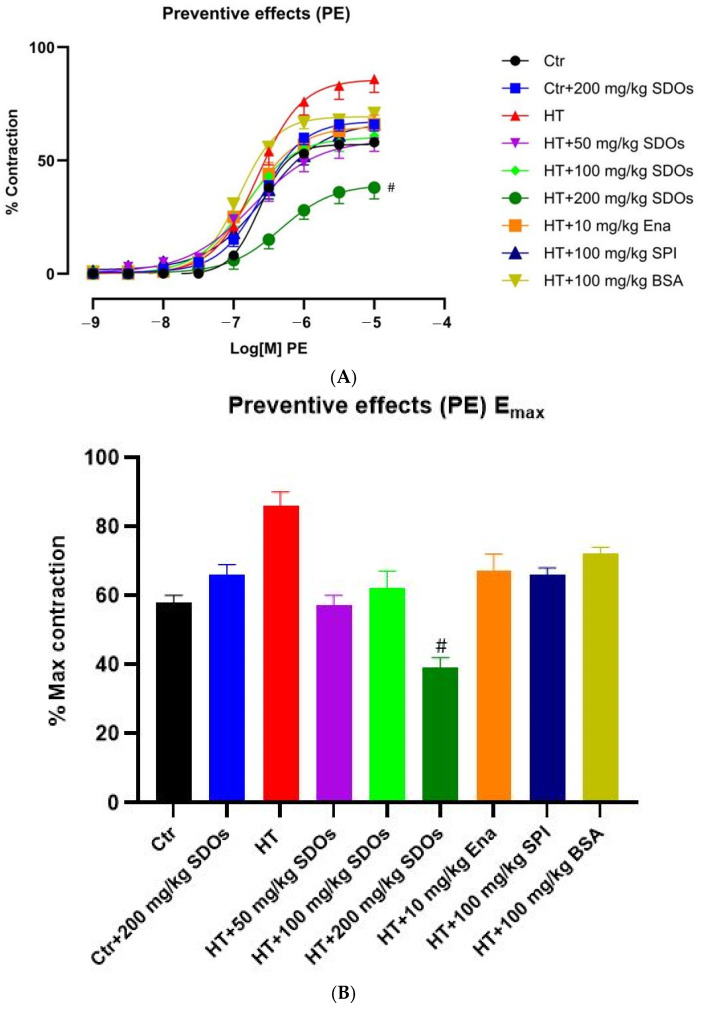
The effects of SDOs on aortic contraction induced with PE (**A**) and the maximum relaxation percentage (**B**) obtained from graph A during the preventive test. # *p* < 0.05 compared to the hypertensive group (HT).

**Figure 10 foods-14-01256-f010:**
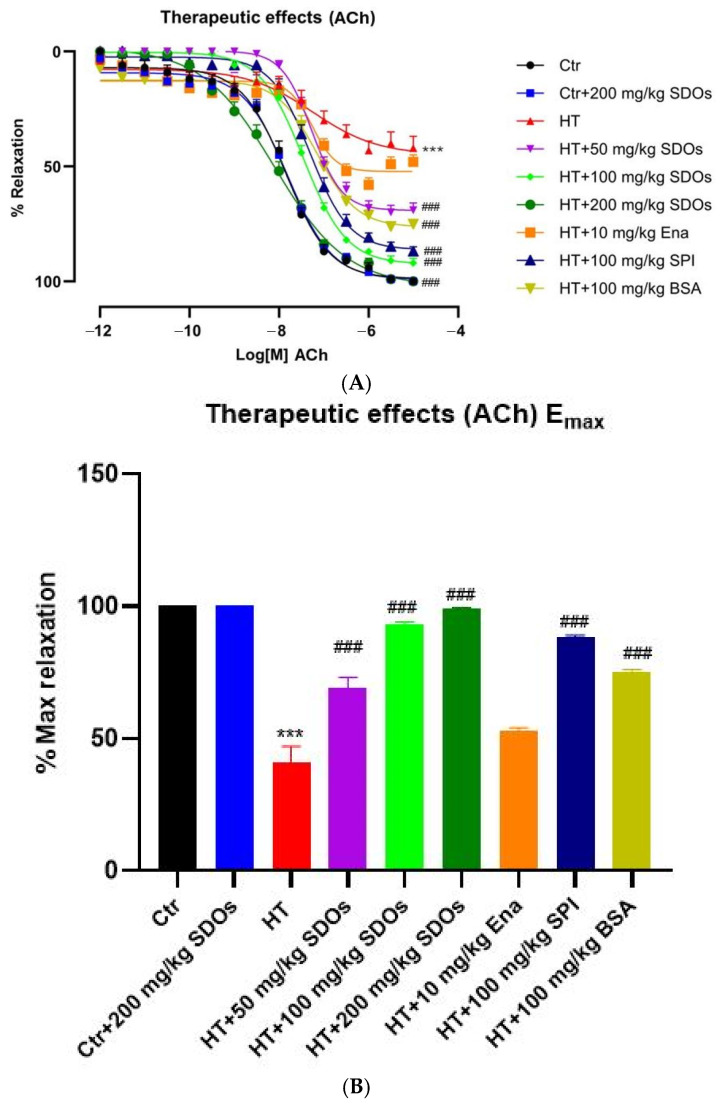
The effects of SDOs on aortic relaxation induced with ACh (**A**) and the maximum relaxation percentage (**B**) obtained from graph A during the therapeutic test. *** *p* < 0.001 compared to the control group (Ctr). ### *p* < 0.001 compared to the hypertensive group (HT).

**Figure 11 foods-14-01256-f011:**
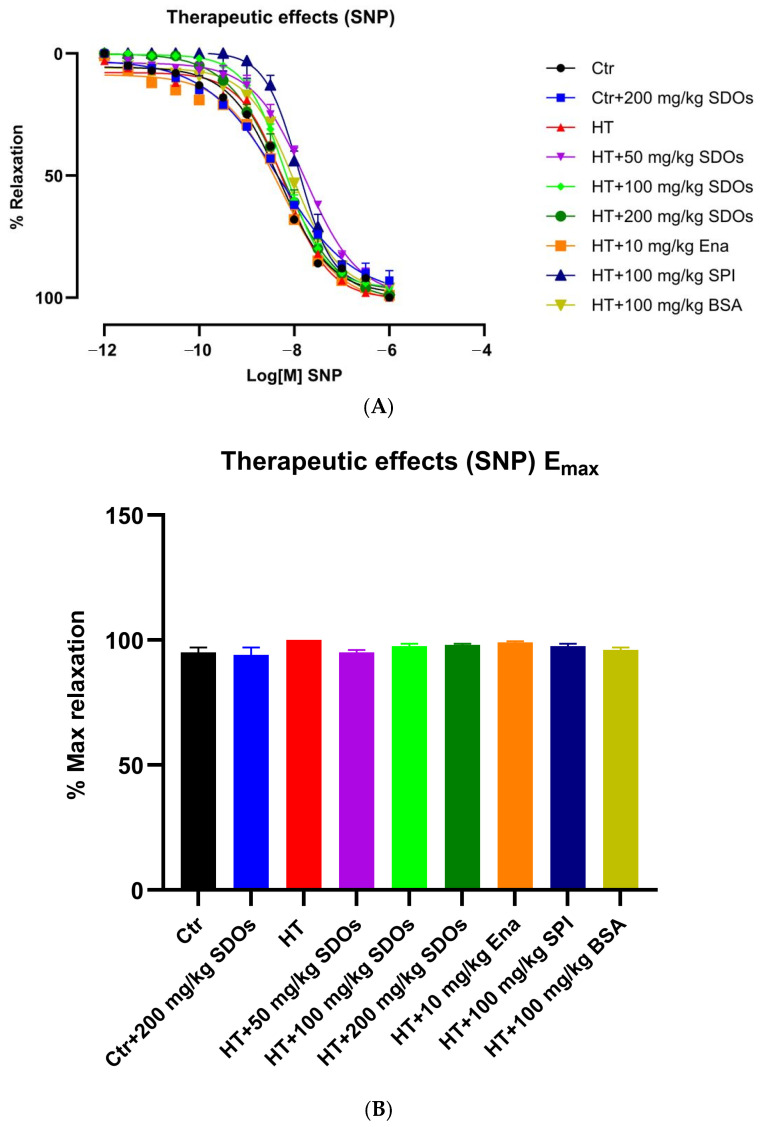
The effects of SDOs on aortic relaxation induced with SNP (**A**) and the maximum relaxation percentage (**B**) obtained from graph A during the therapeutic test.

**Figure 12 foods-14-01256-f012:**
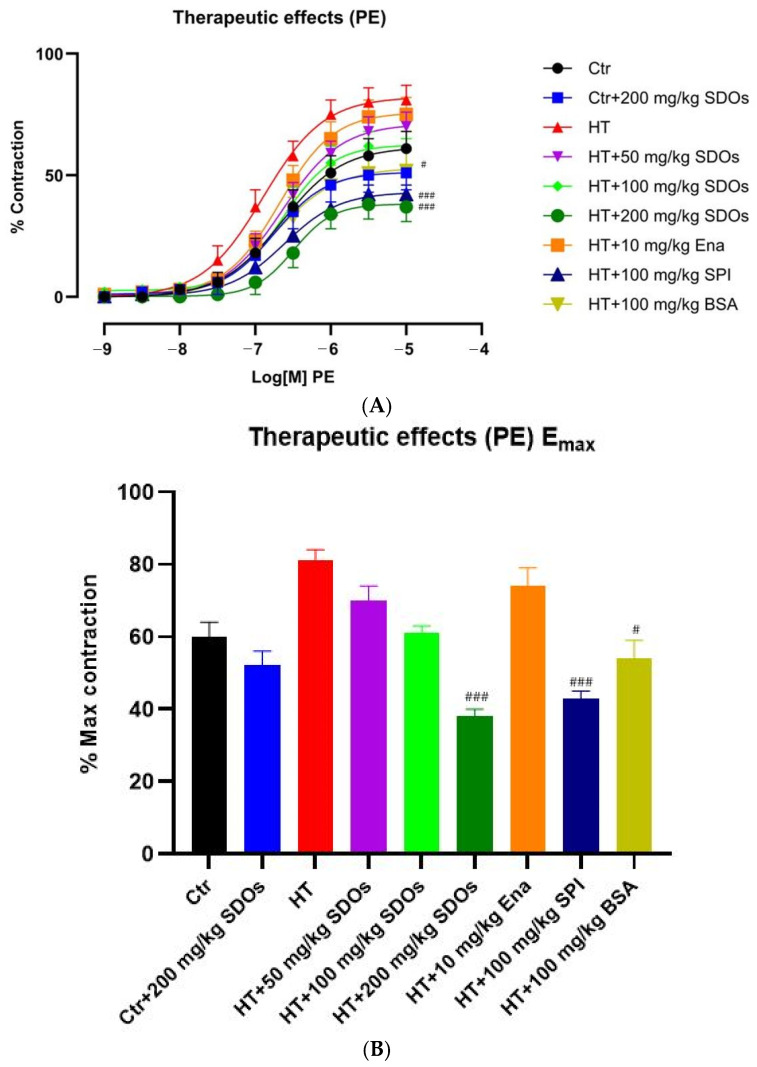
The effects of SDOs on aortic contraction induced with PE (**A**) and the maximum relaxation percentage (**B**) obtained from graph A during the therapeutic test. # *p* < 0.05, ### *p* < 0.001 compared to the hypertensive group (HT).

## Data Availability

The original contributions presented in the study are included in the article; further inquiries can be directed to the corresponding author.

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
