# Peer review of "Preventive and Therapeutic Effects of Sericin-Derived Oligopeptides (SDOs) from Yellow Silk Cocoons on Blood Pressure Lowering in L-NAME-Induced Hypertensive Rats"

_foods, 2025, doi:10.3390/foods14071256_

Round 1
Reviewer 1 Report
Comments and Suggestions for Authors
Dear All,
First, I would like to congratulate you on the excellent work presented in the article. It is highly relevant to the field of functional foods, as it demonstrates an innovative approach by studying preventive and therapeutic effects of SDOs in hypertension. The article is well-written and theoretically grounded, presenting solid and promising results. However, I believe that in order to stay within the scope of a food journal it would be important to characterize SDOs and relate them to the effects observed in hypertension.
Therefore, I suggest that characterization analyses be included, for example: Protein content, amino-acids identification, Peptides’ molecular weight measurement, antioxidant capacity, among others.
Another suggestion is to update the cited references, since most are over 10 years old.
Author Response
|
1. Summary |
|
|
|
Thank you for your time and for your encouraging words regarding our work. I really appreciate your comments and suggestions on the revised text. SDOs have shown the ability to inhibit both ACE and DPP-IV in vitro, corresponding to findings observed in animal studies. The work is part of an investigation initiative researching the development of yellow silk cocoons, historically grown in northeastern Thailand for over a century, as functional dietary components. We have published several research papers that investigate SDOs in more detail, including their protein content, amino acid compositions, molecular peptide weights, active peptide sequences, antioxidant properties, and potential mechanisms of action in vitro, in vivo, and in ex vivo. I will use the information in these materials to respond to your questions.
1.1 Sangsawad, P., Chumee, S., Laosam, P., Roytrakul, S., Sasikan Katemala, S., and Sutheerawattananonda, M. Pilot-Scale Production of Sericin-Derived Oligopeptides (SDOs) from Yellow Silk Cocoons: Peptide Characterization and Specifications. Foods 2025, 14, 500. https://doi.org/10.3390/foods14030500
1.2 Tocharus, C.; Prum, V.; Sutheerawattananonda, M. Oral Toxicity and Hypotensive Influence of Sericin-Derived Oligopeptides (SDOs) from Yellow Silk Cocoons of Bombyx mori in Rodent Studies. Foods 2024, 13, 3505. https://doi.org/10.3390/foods13213505
1.3 Tocharus, C.; Sutheerawattananonda, M. Hypoglycemic Ability of Sericin-Derived Oligopeptides (SDOs) from Bombyx mori Yellow Silk Cocoons and Their Physiological Effects on Streptozotocin (STZ)-Induced Diabetic Rats. Foods 2024, 13, 2184. https://doi.org/10.3390/foods13142184
1.4 Sangsawad, P.; Katemala, S.; Pao, D.; Suwanangul, S.; Jeencham, R.; Sutheerawattananonda, M. Integrated Evaluation of Dual-Functional DPP-IV and ACE Inhibitory Effects of Peptides Derived from Sericin Hydrolysis and Their Stabilities during In Vitro-Simulated Gastrointestinal and Plasmin Digestions. Foods 2022, 11, 3931. https://doi.org/10.3390/foods11233931
1.5 Onsa-Ard, A.; Shimbhu, D.; Tocharus, J.; Sutheerawattananonda, M.; Pantan, R.; Tocharus, C. Hypotensive and vasorelaxant effects of sericin-derived oligopeptides in rats. ISRN Pharmacol. 2013, 2013, 717529. https://doi.org/10.1155/2013/717529
|
||
|
2. Questions for General Evaluation |
Reviewer’s Evaluation |
Response and Revisions |
|
Does the introduction provide sufficient background and include all relevant references? |
Yes |
Improved |
|
Is the research design appropriate? |
Must be improved |
Improved |
|
Are the methods adequately described? |
Yes |
Improved |
|
Are the results clearly presented? |
Must be improved |
Improved |
|
Are the conclusions supported by the results?
|
Yes |
Improved |
|
3. Point-by-point response to Comments and Suggestions for Authors |
||
|
Comments 1: The article is well-written and theoretically grounded, presenting solid and promising results. However, I believe that in order to stay within the scope of a food journal it would be important to characterize SDOs and relate them to the effects observed in hypertension.
|
||
|
Response 1: We added the information to Line 137-141. “The specifications of SDOs derived from yellow silk cocoons were previously detailed in Tocharus et al. (2024). Comprehensive characteristic analyses and stabilities during in vitro-simulated gastrointestinal and plasmin digestions of SDOs have been published and available in Sangsawad et al. (2025) and Sangsawad et al. (2022), respectively.“.
Information regarding the characterization of SDOs is available at https://doi.org/10.3390/foods11233931 and https://doi.org/10.3390/foods14030500. We advanced from the laboratory phase to pilot scale in preparation for commercial production. The study results align with previously published research from our animal experiments, as detailed in the Summary section in reference no. 1.2, 1.3, and 1.5. This research has thoroughly examined various facets of the industrial process, pharmacological studies, and animal research. We are actively pursuing research funding to advance our work, adhere to legal and regulatory standards for innovative food additives, and enhance academic knowledge. After completing each section, we assessed the order for publication. Consequently, the information was presented sequentially, rather than in the order of research completion. This method guarantees the timely dissemination of the most relevant findings, regardless of their completion dates. Publishing in a logical sequence enhances the clarity and impact of research contributions.
|
||
|
Comments 2: I suggest that characterization analyses be included, for example: Protein content, amino-acids identification, Peptides’ molecular weight measurement, antioxidant capacity, among others.
|
||
|
Response 2: Our previously published research, which is available at https://doi.org/10.3390/foods14030500, includes detailed characterization analyses of protein content, amino acid identification, molecular weight of peptides, analyses of ABTS Radical Scavenging Activity, Ferric Reducing Antioxidant Power (FRAP), and Metal Chelating Activity, as well as a shelf-life study based on bioactivity for ACE and DPP-IV inhibition's bioactivity (Figure 5 in the paper).
|
||
|
Figure 5. Schematic overview of SDO production process and characterization.
Comments 3: Another suggestion is to update the cited references, since most are over 10 years old.
|
||
|
Response 3: We revised the referenced sources as indicated but kept the originals since we wanted to credit such important concepts. This article also comprises more than 20-year-old research on how BSA and SPI control blood pressure by testing for ACE inhibitory activity and functional vascular testing. The paper used both old and new studies to show the possible benefits of mixing protein-rich foods from different sources to meet customer needs while keeping their bioactive properties.
|
||

Reviewer 2 Report
Comments and Suggestions for Authors
What is the main question addressed by the research?
Present study addressed very interesting question of the effects of bioactive peptides (Sericin-Derived Oligo-2 peptides) from Yellow Silk Cocoons on Blood Pressure lowering in L-NAME-Induced hypertensive rats. The authors emphasize the potential positive preventive and therapeutic effect in a hypertension model.
Do you consider the topic original or relevant in the field? Does it
address a specific gap in the field?
The topic the authors address is original and current. However, the manuscript has serious shortcomings that need to be corrected, as well as additional experiments to make the study more complete.
- The abstract is very poorly written. It is confusing and does not fully convey the essence of the text. The abstract should clearly specify the groups that are included in this paper. From the abstract it can be concluded that there are 3-4 groups in the paper, and there are 9 of them.
- Too long and unnecessary discussion about hypertension and presenting too many facts that are not closely related to the study, which distracts the reader from the focus of the topic.
- In the introduction, better describe and explain why the YPDLPYH sequence is important for the study.
- Line 58:” The ex vivo study demonstrated that SDOs lower blood pressure via mechanisms involving smooth muscle vasodilation and calcium ion channel blockade” . In this study, they use enalapril as a positive control. Enalapril is ACE inhibitor. If they already state that SDO acts as a calcium ion channel blocker, why wasn't there a group with a substance from that group of drugs?
- Line 69: The authors mention cholesterol levels and the effect of SDO on cholesterol. What does that have to do with this study?
- Line 75: “Considering the previously discussed health advantages of SDOs from yellow silk cocoons, several research have shown their simultaneous effects on multiple organs [9,17,18].” Rephrase the sentence because references 9,17 and 18 belong to the same group of authors as this manuscript. It may mislead the reader into thinking that the research in question is by another group of authors. Start with "our previous research..." I think it is unethical to write it this way.
- Line 80:” They do not only provide amino acids similar to dietary protein but also exhibit significant antioxidant properties, possibly enhancing the overall health and fitness of patients with hypertension.” There are no antioxidant studies in this manuscript. Either supplement with results or remove this type of speculation from this manuscript.
- In the AIM of this study, SPI and BSA appear suddenly, without prior notice. In the introduction, explain why these substances were used in the experiment.
- Why Enalapril and not Verapamil?
- Figure one shows six groups and it says there are 9. Clarify the picture and explain that in SDOs groups there are four groups, one of which does not receive Lname. Explain better why one group only receives SDO.
- Authors used one-way analysis of variance (ANOVA) for statistical data 213 analysis and utilized the LSD test to assess group differences. The statistics are very questionable. I suggest comparing the groups: control group with control group + 200SDO and control group with Hypertension group. The first comparison confirms that there is no effect on healthy animals, the second comparison confirms that the hypertension model is established. Groups 3-9 should be compared by ANOVA. In the graphs in the manuscript, the difference exists only between the control and the hypertensive groups. Such results are unlikely if ANOVA was performed. There must also be a difference between the treated groups.
- What does aging and mitochondrial dysfunction have to do with this manuscript? The animals are of the same age, and mitochondrial dysfunction was not proven nor addressed in the results.
- Line 364: Two peptide sequences with the highest inhibitory for ACE were CEF (Cysteine-Glutamic acid-Phenylalanine) and YPDLPYH (Tyrosine-Proline-Aspartic acid-Leucine-Proline-Tyrosine-Histidine). Why were these sequences not proven in this study by some molecular biological methods?
- Better arrange abbreviations in the text.
- Due to the results with SNP, it is necessary to perform a group with a positive control of NO donor. Recommendation to perform with L arginine
- Lines 610-613 are repeated from the introduction; they are not necessary.
- 609-631 I don't understand the significance of the text for this experimental work. the general facts of the action of drugs, which were not even tested within this work, are explained.
- 633-649 I don't understand the significance of the text for this experimental work. why these statements are specifically important for explaining these results obtained in the experiment.
- 650-658 I don't understand the significance of the text for this experimental work. why these statements are specifically important for explaining these results obtained in the experiment.
- 659-664 I don't understand the significance of the text for this experimental work. there is no connection with the substance being tested.
- In Lname-induced hypertension, kidney damage occurs after 4 weeks with such high doses of Lname, I recommend that the authors publish the results of proteinuria, albuminuria and histopathological images of the kidneys, because they often repeat nephroprotective effects in the text.
The English language needs to be edited. Native speaker help is needed.
Author Response
|
1. Summary |
|
|
|
Thank you for your time and inquiries about our work. I appreciate your comments and ideas on the revised text, particularly the lines "poorly written" and "I don't understand," which prompted us to go over the document. Thank you for being honest and forthright in your evaluation of the manuscript's content. This significantly improved the updated text. Previous study has demonstrated that SDOs derived from yellow silk cocoons may inhibit both ACE and DPP-IV in vitro, which is consistent with results from animal trials. The experiment is part of an investigation into the development of yellow silk cocoons, which have been farmed in northeastern Thailand for more than a century, as useful dietary components. We have published numerous research publications that look into SDOs in further depth, including their protein content, amino acid compositions, molecular peptide weights, active peptide sequences, antioxidant characteristics, and probable modes of action in vitro, in vivo, and ex vivo. I will utilize the information in these documents to answer your queries.
1.1 Sangsawad, P., Chumee, S., Laosam, P., Roytrakul, S., Sasikan Katemala, S., and Sutheerawattananonda, M. Pilot-Scale Production of Sericin-Derived Oligopeptides (SDOs) from Yellow Silk Cocoons: Peptide Characterization and Specifications. Foods 2025, 14, 500. https://doi.org/10.3390/foods14030500
1.2 Tocharus, C.; Prum, V.; Sutheerawattananonda, M. Oral Toxicity and Hypotensive Influence of Sericin-Derived Oligopeptides (SDOs) from Yellow Silk Cocoons of Bombyx mori in Rodent Studies. Foods 2024, 13, 3505. https://doi.org/10.3390/foods13213505
1.3 Tocharus, C.; Sutheerawattananonda, M. Hypoglycemic Ability of Sericin-Derived Oligopeptides (SDOs) from Bombyx mori Yellow Silk Cocoons and Their Physiological Effects on Streptozotocin (STZ)-Induced Diabetic Rats. Foods 2024, 13, 2184. https://doi.org/10.3390/foods13142184
1.4 Sangsawad, P.; Katemala, S.; Pao, D.; Suwanangul, S.; Jeencham, R.; Sutheerawattananonda, M. Integrated Evaluation of Dual-Functional DPP-IV and ACE Inhibitory Effects of Peptides Derived from Sericin Hydrolysis and Their Stabilities during In Vitro-Simulated Gastrointestinal and Plasmin Digestions. Foods 2022, 11, 3931. https://doi.org/10.3390/foods11233931
1.5 Onsa-Ard, A.; Shimbhu, D.; Tocharus, J.; Sutheerawattananonda, M.; Pantan, R.; Tocharus, C. Hypotensive and vasorelaxant effects of sericin-derived oligopeptides in rats. ISRN Pharmacol. 2013, 2013, 717529. https://doi.org/10.1155/2013/717529
|
||
|
2. Questions for General Evaluation |
Reviewer’s Evaluation |
Response and Revisions |
|
Does the introduction provide sufficient background and include all relevant references? |
Must be improved |
Improved |
|
Is the research design appropriate? |
Can be improved |
Improved |
|
Are the methods adequately described? |
Can be improved |
Improved |
|
Are the results clearly presented? |
Must be improved |
Improved |
|
Are the conclusions supported by the results? |
Must be improved |
Improved |
|
3. Point-by-point response to Comments and Suggestions for Authors |
||
|
Comments 1: What is the main question addressed by the research? Present study addressed very interesting question of the effects of bioactive peptides (Sericin-Derived Oligo-2 peptides) from Yellow Silk Cocoons on Blood Pressure lowering in L-NAME-Induced hypertensive rats. The authors emphasize the potential positive preventive and therapeutic effect in a hypertension model.
|
||
|
Response 1: The key question for this study originated from our prior findings, which showed that SDOs inhibited both ACE and DPP-IV in vitro, as well as decreasing blood pressure in normotensive rats during oral toxicity testing. Following that, we investigated whether SDOs may have therapeutic or preventive benefits when L-NAME was present, as opposed to other dietary proteins such as SPI and BSA, using enalapril as a control.
|
||
|
Comments 2: Do you consider the topic original or relevant in the field? |
||
|
Response 2: Original. Few research studies have specifically reported on SDOs from yellow silk cocoons, which differ in morphology and chemical composition from white silk cocoons. Of particular interest is the ability of the silkworm to transport lutein from mulberry leaves, forming a structure that is only proven to be similar to the lutein present in the human retina. Yellow silk cocoons are considered inferior to the white cocoons for the textile industry as they have short silk yarn and a high content of sericin that needs to be removed with wastewater. However, yellow silk cocoons can be advantageous in certain applications due to their unique properties, which may be beneficial for specific textiles or eco-friendly practices. Additionally, the higher sericin content in yellow cocoons could offer potential health benefits and opportunities for innovation in cosmetic and biomedical fields. Based on our results from several investigations, yellow silk cocoons can be a good candidate for a functional food ingredient.
Relevant. The study aims and techniques are similar to those used in other peptide investigations, beginning with determining the best digesting conditions to produce high bioactive peptides, peptide characterization analyses, stability in the GI and blood circulation systems, and effectiveness in animal experiments. We have periodically published some of our work in references 1.1-1.5.
|
||
|
Comments 3: Does it address a specific gap in the field? |
||
|
Response 3: We have added the specific gap for this study as follows: However, researchers have yet to explore their ability to prevent and treat high blood pressure in rats, as well as their effects on vascular function under hypertensive conditions. |
||
|
Comments 4: The topic the authors address is original and current. However, the manuscript has serious shortcomings that need to be corrected, as well as additional experiments to make the study more complete. |
||
|
Response 4: The manuscript has been revised in accordance with your suggestions. All the relevant information for this investigation is available in our previously published work (references 1.1-1.5), as noted in the text. This includes SDOs, peptide analysis and characterization, SDO requirements, stability in the simulated gastrointestinal system and blood circulation, shelf-life testing, active peptide sequences, and potential pathways of action. The findings illustrate the comprehensiveness of the study and its dependence on prior results. This indicates that our current work is adequately supported and grounded in established evidence.
|
||
|
Comments 5: The abstract is very poorly written. It is confusing and does not fully convey the essence of the text. The abstract should clearly specify the groups that are included in this paper. From the abstract it can be concluded that there are 3-4 groups in the paper, and there are 9 of them.
|
||
|
Response 5: We added specific, detailed information about the number of rat groups as follows: The experiment involved nine rat groups: 1) Normal control, 2) Normal + 200 BW SDOs, 3) Hypertensive (HT) control, 4) HT + 50 BW SDOs, 5) HT + 100 BW SDOs, 6) HT + 200 BW SDOs, 7) HT + enalapril, 8) HT + soy protein isolate (SPI), and 9) HT + bovine serum albumin (BSA).
|
||
|
Comments 6: Too long and unnecessary discussion about hypertension and presenting too many facts that are not closely related to the study, which distracts the reader from the focus of the topic.
|
||
|
Response 6: The manuscript has undergone significant revisions in accordance with your recommendations.
|
||
|
Comments 7: In the introduction, better describe and explain why the YPDLPYH sequence is important for the study. |
||
|
Response 7: The information was deleted, and a reference was provided for this particular data.
|
||
|
Comments 8: Line 58:” The ex vivo study demonstrated that SDOs lower blood pressure via mechanisms involving smooth muscle vasodilation and calcium ion channel blockade”. In this study, they use enalapril as a positive control. Enalapril is ACE inhibitor. If they already state that SDO acts as a calcium ion channel blocker, why wasn't there a group with a substance from that group of drugs?
|
||
|
Response 8: SDOs reduce blood via different mechanisms, according to our earlier studies. Strong ACE inhibitory properties, resistance to simulated GI and blood protease digestion, high antioxidant activity, smooth muscle vasodilation, and calcium ion channel blocking are some of these mechanisms. This research compared the findings to SPI and BSA, which have antioxidant and ACE inhibitory properties. Hypertensives suggest enalapril initially. If kidney function deteriorates, doctors recommend a calcium ion channel blocker antihypertensive medication, which is more expensive than ACE inhibitors, especially in Thailand.
|
||
|
Comments 9: Line 69: The authors mention cholesterol levels and the effect of SDO on cholesterol. What does that have to do with this study?
|
||
|
Response 9: We referenced our prior studies to demonstrate the reduced risk of cardiovascular diseases. We removed this information in the updated text.
|
||
|
Comments 10: Line 75: “Considering the previously discussed health advantages of SDOs from yellow silk cocoons, several research have shown their simultaneous effects on multiple organs [9,17,18].” Rephrase the sentence because references 9,17 and 18 belong to the same group of authors as this manuscript. It may mislead the reader into thinking that the research in question is by another group of authors. Start with "our previous research..." I think it is unethical to write it this way. |
||
|
Response 10: The sentence was rephrased according to your recommendation. |
||
|
Comments 11: Line 80:” They do not only provide amino acids similar to dietary protein but also exhibit significant antioxidant properties, possibly enhancing the overall health and fitness of patients with hypertension.” There are no antioxidant studies in this manuscript. Either supplement with results or remove this type of speculation from this manuscript.
|
||
|
Response 11: We have already published this information in section 1.1. A reference was included to substantiate this information.
1.1 Sangsawad, P., Chumee, S., Laosam, P., Roytrakul, S., Sasikan Katemala, S., and Sutheerawattananonda, M. Pilot-Scale Production of Sericin-Derived Oligopeptides (SDOs) from Yellow Silk Cocoons: Peptide Characterization and Specifications. Foods 2025, 14, 500. https://doi.org/10.3390/foods14030500
|
||
|
Comments 12: |
||
|
Response 12: We revised the introductory part for SPI and BSA to include it in the introduction.
|
||
|
Comments 13: Why Enalapril and not Verapamil? |
||
|
Response 13: While it is an excellent concept, as previously addressed in Comment 8, SDOs possess several pathways for blood-lowering efficacy to allow comparisons with proteins exhibiting similar characteristics and possible practical uses. We chose to use enalapril as a positive control.
|
||
|
Comments 14: Figure one shows six groups and it says there are 9. Clarify the picture and explain that in SDOs groups there are four groups, one of which does not receive Lname. Explain better why one group only receives SDO. |
||
|
Response 14: We have changed the figure in accordance with your suggestion.
|
||
|
Comments 15: Authors used one-way analysis of variance (ANOVA) for statistical data 213 analysis and utilized the LSD test to assess group differences. The statistics are very questionable. I suggest comparing the groups: control group with control group +200 SDOs and control group with Hypertension group. The first comparison confirms that there is no effect on healthy animals, the second comparison confirms that the hypertension model is established. Groups 3-9 should be compared by ANOVA. In the graphs in the manuscript, the difference exists only between the control and the hypertensive groups. Such results are unlikely if ANOVA was performed. There must also be a difference between the treated groups. |
||
|
Response 15: We have reanalyzed the data and changed significant symbols in the respective plots, including the text in Section 2.5.
|
||
|
Comments 16: What does aging and mitochondrial dysfunction have to do with this manuscript? The animals are of the same age, and mitochondrial dysfunction was not proven nor addressed in the results. |
||
|
Response 16: This information has been removed.
|
||
|
Comments 17: Line 364: Two peptide sequences with the highest inhibitory for ACE were CEF (Cysteine-Glutamic acid-Phenylalanine) and YPDLPYH (Tyrosine-Proline-Aspartic acid-Leucine-Proline-Tyrosine-Histidine). Why were these sequences not proven in this study by some molecular biological methods? |
||
|
Response 17: This information has been removed since full data has previously been published in Sangsawad et al. (2025).
|
||
|
Comments 18: Better arrange abbreviations in the text. |
||
|
Response 18: We changed them in accordance with your recommendation. |
||
|
Comments 19: Due to the results with SNP, it is necessary to perform a group with a positive control of NO donor. Recommendation to perform with L arginine |
||
|
Response 19: According to the supplementary figure (Figure 4), the action site of acetylcholine must activate endothelial nitric oxide synthase (eNOS) in endothelial cells to its active state, allowing L-arginine to react with eNOS to produce nitric oxide (NO) in the presence of oxygen (O₂). Nonetheless, SNP is a nitric oxide donor that may directly interact with smooth muscle cells. L-NAME decreased the total number of active eNOS forms, leading to a reduction in NO production in hypertensive rats (Panthiya et al., 2021). We understand that L-arginine would produce effects comparable to acetylcholine, since it requires interaction with endothelial cells, unlike sodium nitroprusside, which directly influences smooth muscle cells to confirm their sustained functionality throughout all experimental groups. Luckika Panthiya, Jiraporn Tocharus , Amnart Onsa-ard, Waraluck Chaichompoo, Apichart Suksamrarn, Chainarong Tocharus . Hexahydrocurcumin ameliorates hypertensive and vascular remodeling in L-NAME-induced rats. BBA - Molecular Basis of Disease 1868 (2022) 166317 https://doi.org/10.1016/j.bbadis.2021.166317 |
||
|
Comments 20: Lines 610-613 are repeated from the introduction; they are not necessary. |
||
|
Response 20: Following your suggestion, we have removed this section.
|
||
|
Comments 21: 609-631 I don't understand the significance of the text for this experimental work. the general facts of the action of drugs, which were not even tested within this work, are explained. |
||
|
Response 21: Following your suggestion, we have removed this section.
|
||
|
Comments 22: 633-649 I don't understand the significance of the text for this experimental work. why these statements are specifically important for explaining these results obtained in the experiment. |
||
|
Response 22: Following your suggestion, we have removed this section.
|
||
|
Comments 23: 650-658 I don't understand the significance of the text for this experimental work. why these statements are specifically important for explaining these results obtained in the experiment. |
||
|
Response 23: Following your suggestion, we have removed this section.
|
||
|
Comments 24: 659-664 I don't understand the significance of the text for this experimental work. there is no connection with the substance being tested. |
||
|
Response 24: Following your suggestion, we have removed this section.
|
||
|
Comments 25: In Lname-induced hypertension, kidney damage occurs after 4 weeks with such high doses of Lname, I recommend that the authors publish the results of proteinuria, albuminuria and histopathological images of the kidneys, because they often repeat nephroprotective effects in the text. |
||
|
Response 25: I appreciate your insightful recommendation once again. This research focused on hypertension and vascular activity after we completed our toxicology experiment some years ago. We postponed publication of this work to allow for the completion of more research studies on SDOs to confirm our results. In the future, we will be focusing on the advantages of SDOs for renal health.
|
||
|
4. Response to Comments on the Quality of English Language |
||
|
Point 1: The English language needs to be edited. Native speaker help is needed. |
||
|
Response 1: We will proceed according to your advice.
|
||

Reviewer 3 Report
Comments and Suggestions for Authors
Dear Authors,
The manuscript with title "Preventive and Therapeutic Effects of Sericin-Derived Oligopeptides (SDOs) from Yellow Silk Cocoons on Blood Pressure Lowering in L-NAME-Induced Hypertensive Rats" is very interesting and deals with important results which can be applicable in pharmaceutical purposes since the hypertension is one of the most abundant cardiovascular diseases.
However, this manuscript should be improved in several points:
- Line 192-193: The authors stated "The blood vessels were submerged in Krebs's solution, which included (mM): NaCl; KCl 5; [N-(2-hydroxy-ethyl) piperazine N-(2-ethanesulfonic acid)]." The authors should explain more precisely which is the effect of Sericin-Derived Oligopeptides on the concentration of K+. The explanation is very short in line 241-244.
- Line 204-206: The authors should explain why phenylephrine, acetylcholine and sodium nitroprusside were the best choice to evaluate the functionality of blood arteries?
- Which temperature was applied in organic bath?
- Would be the similar effect of oligopeptides on Ca2+ and other stimulants such caffeine, tannins etc.
- The authors should explain are the proposed dosages 100 and 200 mg kg⁻¹ BW (noticed in conclusion) are safe regarding the toxic effect of Sericin-Derived Oligopeptides on liver (hepatotoxicity).
I suggest major revision of the manuscript.
Author Response
|
1. Summary |
|
|
|
We appreciate your time and attention to specific details that were unclear, which greatly contributed to the improvement of the revised manuscript. We have incorporated specific details, and a figure based on your recommendations to enhance the reader's understanding of our work. The revised manuscript provides a point-by-point response in accordance with your suggestions. Your comments and specific ideas, such as the roles of SDOs on Ca²⁺ and K⁺ channels, are greatly appreciated. This has clarified SDO's mechanisms of action related to hypotensive effects and vascular activity.
|
||
|
2. Questions for General Evaluation |
Reviewer’s Evaluation |
Response and Revisions |
|
Does the introduction provide sufficient background and include all relevant references? |
Yes |
Improved |
|
Is the research design appropriate? |
Can be improved |
Improved |
|
Are the methods adequately described? |
Can be improved |
Improved |
|
Are the results clearly presented? |
Yes |
Improved |
|
Are the conclusions supported by the results?
|
Yes |
Improved |
|
3. Point-by-point response to Comments and Suggestions for Authors |
||
|
Comments 1: Line 192-193: The authors stated "The blood vessels were submerged in Krebs's solution, which included (mM): NaCl; KCl 5; [N-(2-hydroxy-ethyl) piperazine N-(2-ethanesulfonic acid)]." The authors should explain more precisely which is the effect of Sericin-Derived Oligopeptides on the concentration of K+. The explanation is very short in line 241-244. |
||
|
Response 1: In the highlight section, we discussed how L-NAME induces hypertension. However, the results from our earlier work, Onsa-ard et al. (2013), may explain the influence of SDOs on K+ concentrations. This study used endothelium-denuded aortic rings from normotensive rats to investigate vasorelaxation mechanisms independent of the endothelium. They were preincubated with various K+ channel blockers before PE precontraction, and the relaxation curve was measured in response to increasing SDO concentrations. Except for tetraethylammonium (a Ca²⁺-activated K⁺ channel blocker), all examined K⁺ channel blockers showed no influence on the concentration-response curve of SDOs. SDOs' relaxing action may contribute to the opening of Ca²⁺-activated K⁺ channels in vascular smooth muscle cells. |
||
|
Onsa-Ard, A.; Shimbhu, D.; Tocharus, J.; Sutheerawattananonda, M.; Pantan, R.; Tocharus, C. Hypotensive and vasorelaxant effects of sericin-derived oligopeptides in rats. ISRN Pharmacol. 2013, 2013, 717529. https://doi.org/10.1155/2013/717529
Comments 2: Line 204-206: The authors should explain why phenylephrine, acetylcholine and sodium nitroprusside were the best choice to evaluate the functionality of blood arteries?
|
||
|
Response 2: We included information from Lines 238-252 to 2.4 Functional Vascular Study and Figure 4 to demonstrate the action sites of phenylephrine (PE), acetylcholine (ACh), and sodium nitroprusside (SNP) and to investigate how effective blood vessels function via EC and SMC.
|
||
|
Comments 3: Which temperature was applied in organic bath? |
||
|
Response 3: We maintained the organ bath temperature at 37°C under oxygenated conditions. In accordance with your recommendation, we have included this information in section 2.4 Functional Vascular Study Line 222-225, which aligns with the details shown in Figure 3 of the submitted publication.
|
||
|
Comments 4: Would be the similar effect of oligopeptides on Ca2+ and other stimulants such caffeine, tannins etc. |
||
|
Response 4: In Onsa-Ard et al. (2013), endothelium-denuded rings were preincubated in KCl and bathed in Ca²⁺-free Kreb's solution containing 1 mM EGTA to study the function of SDOs in Ca²⁺ release from intracellular storage sensitive to PE and caffeine. SDOs significantly reduced PE-induced contractions, which initiate IP₃-dependent Ca²⁺ release from the intracellular storage. Caffeine-induced contractions, which release Ca²⁺ from intracellular reserves independently of IP₃, were unchanged. SDOs may reduce IP₃-dependent Ca²⁺ releases from SR sensitive to PE, contributing to their vascular effects. Additionally, caffeine (10 mM) has been used in various publications to deplete the SR of Ca²⁺ from muscle tissues. This suggests that while SDOs can inhibit the specific pathway activated by PE, they do not affect the alternative mechanism of Ca²⁺ release triggered by caffeine. So, the different responses to these agents show that SDOs specifically target IP3 -mediated processes.
Tannin may cause muscular contraction as well as relaxation. However, it can only cause contraction of intact endothelial aortic rings, not endothelial-denuded rings. This process is connected to the release of chemicals from the EC that make muscles contract and relax. For example, thromboxane A2 makes muscles contract, and NO makes muscles relax (Russel and Rohrbach, 1989). This difference emphasizes the role of endothelial cells in mediating tannic effects, since their presence is required for the release of specific factors that regulate muscle tone. As a result, the endothelium plays an important role in regulating how tannin affects vascular smooth muscles.
SDOs have a direct effect on both endothelium and vascular smooth muscle, leading to vasodilation. The results suggested that SDOs have a dose-dependent relaxing effect on the isolated rat aorta. The relaxing effect of SDOs is mediated through Ca²⁺ antagonism and the NO/sGC/cGMP pathway, which leads to the lowering effect on SBP.
Russel, J.A. and Rohrbach, M.S. Tannin Induces Endothelium-dependent Contraction and Relaxation of Rabbit Pulmonary Artery. AM REV RESPIR DIS 1989; 139:498-503.
|
||
|
Onsa-Ard, A.; Shimbhu, D.; Tocharus, J.; Sutheerawattananonda, M.; Pantan, R.; Tocharus, C. Hypotensive and vasorelaxant effects of sericin-derived oligopeptides in rats. ISRN Pharmacol. 2013, 2013, 717529. https://doi.org/10.1155/2013/717529
Comments 5: The authors should explain are the proposed dosages 100 and 200 mg kg⁻¹ BW (noticed in conclusion) are safe regarding the toxic effect of Sericin-Derived Oligopeptides on liver (hepatotoxicity). |
||
|
Response 5: Based on our results that have been previously published in detail (Tocharus et al., 2024), SDOs illustrated safety in both acute and chronic toxicity tests as stated in our abstract: “Clinical chemistry, hematology, and histopathological studies revealed that SDOs were safe for a single dose of 2000 mg kg⁻¹ body weight (BW) and daily oral administration of 50, 100, and 200 mg kg⁻¹ BW for six months. The chronic toxicity study additionally measured the rats’ systolic blood pressure (SBP) and blood sugar monthly as they slowly aged. In the 2nd month for male rats and the 4th month for both sexes, SDOs had a significant hypotensive effect on Wistar rats’ blood pressure, lowering it from 130 mmHg to a plateau at 110–115 mmHg.”
Tocharus, C.; Prum, V.; Sutheerawattananonda, M. Oral Toxicity and Hypotensive Influence of Sericin-Derived Oligopeptides (SDOs) from Yellow Silk Cocoons of Bombyx mori in Rodent Studies. Foods 2024, 13, 3505. https://doi.org/10.3390/foods13213505,
To make it clearer, we revised the conclusions to be as follows:
|
||

Round 2
Reviewer 1 Report
Comments and Suggestions for Authors
As the corrections were made by the authors, the article is in compliance to be accepted.
Reviewer 3 Report
Comments and Suggestions for Authors
Dear Authors,
The manuscript with title "Preventive and Therapeutic Effects of Sericin-Derived Oligopeptides (SDOs) from Yellow Silk Cocoons on Blood Pressure Lowering in L-NAME-Induced Hypertensive Rats" is excellent revised and all the questions are answered completely and accurate. Newly added references are suitable and relevant to the topic. The revised manuscript is with high-quality and one of the rarely completely revised manuscripts.
I strongly suggest acceptance of the revised version of the manuscript.